# Population-based tract-to-region connectome of the human brain and its hierarchical topology

Fang-Cheng Yeh [1,2] ✉

Connectome maps region-to-region connectivities but does not inform which white matter pathways form the connections. Here we constructed a population-based tract-to-region connectome to fill this information gap. The constructed connectome quantifies the population probability of a white matter tract innervating a cortical region. The results show that ~85% of the tract-to-region connectome entries are consistent across individuals, whereas the remaining (~15%) have substantial individual differences requiring individualized mapping. Further hierarchical clustering on cortical regions revealed dorsal, ventral, and limbic networks based on the tract-to-region connective patterns. The clustering results on white matter bundles revealed the categorization of fiber bundle systems in the association pathways. This tract-to-region connectome provides insights into the connective topology between cortical regions and white matter bundles. The derived hierarchical relation further offers a categorization of gray and white matter structures.

Mapping the human connectome is the key to understanding how brain structure gives rise to functions and how brain diseases cause dysfunctions[1,2]. Studies have used structural or functional connectivity to quantify the region-to-region connectivity as the connectome[3,4] and delineate the network topology of the nervous system. The network topology revealed by the brain connectome further informed the functional implications of cortical regions and enabled graphical theoretical analysis[5]. However, the conventional region-to-region connectome is agnostic of the role played by white matter pathways and does not indicate which pathways form the cortical connections. Consequently, for many neuroscience studies investigating region-to-region connectivity, the white matter is still a black box with much unknown that needs further exploration.

Here we mapped the tract-to-region connectome to address this information gap. The connection probability between white matter pathways and cortical regions was evaluated on 1065 young adult subjects. For *m* brain regions and *n* white matter bundles, the tract-to-region connectome can be quantified by an *m*-by-*n* matrix, where each matrix entry records the corresponding population probability of a white matter pathway innervating a cortical region.

Several technical advances were used to construct the tract-to-region connectome (Fig. 1). The white matter bundles of the 1065 young adults were mapped using automated tractography. Although many automated tractography methods are available[6–11], most have used cortical parcellation to recognize tracts. These region-based methods could lead to circular analysis in the tract-to-region connectome. Therefore, this study used trajectory-based recognition[12] and did not filter tractogram by brain regions. After trajectory-based recognition, connections substantially deviated from the expert-vetted tractography atlas were removed.

Four tract-to-region connectome matrices were quantified using the Brodmann parcellation, Kleist parcellation[13], the Human Connectome Project's multimodal parcellations (HCP-MMP)[14], and a random parcellation, respectively. The tract-to-region information provided by this approach could complete the circuit diagram for many structure-function models and inform the likelihood of a white matter lesion causing a functional deficit in the dysconnectome studies. Further hierarchical clustering was applied to the tract-to-region connectome. The clustering results revealed the hierarchical relation of cortical regions and white matter pathways that informed their categorization.

[1]Department of Neurological Surgery, University of Pittsburgh, Pittsburgh, PA, USA. [2]Department of Bioengineering, University of Pittsburgh, Pittsburgh, PA, USA.
✉e-mail: frank.yeh@pitt.edu

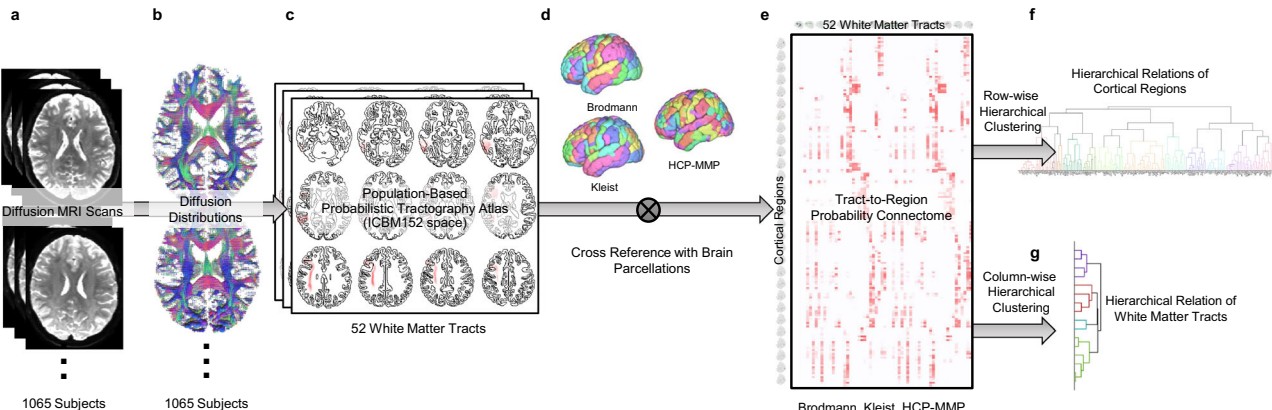

**Fig. 1 | The processing flow to construct a population-based tractography connectome and derive its hierarchical relation. a** The diffusion MRI data of 1065 subjects were used. **b** The data were reconstructed to calculate the diffusion distribution for fiber tracking. **c** For each subject, 52 white matter bundles were mapped using automated tractography. The track recognition was based on trajectory similarity with a tractography atlas without using the cortical parcellations. The tracking results were aggregated to construct a population-based probability atlas of 52 white matter pathways. **d** Cortical regions from cortical parcellations and the white matter trajectories of each subject were used to derive the connectome matrix. **e** The results from each subject were accumulated to construct the tract-region connectome based on population probability. **f** Hierarchical clustering was applied to the row vectors of the connectome to derive the hierarchical relation of cortical regions. **g** Hierarchical clustering was applied to the column vectors to derive the hierarchical relation of white matter bundles.

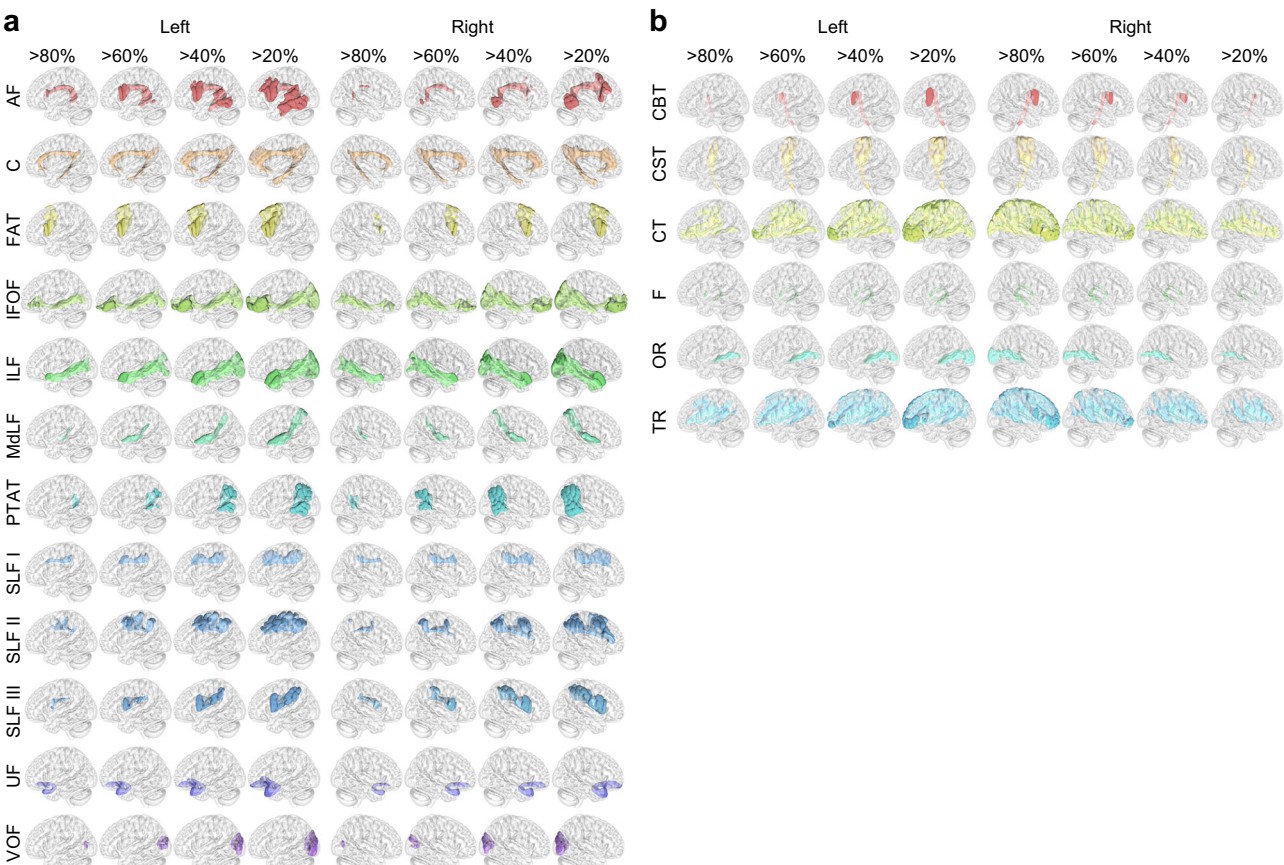

**Fig. 2 | Probabilistic tractography atlas of white matter pathways visualized at population probability of 20, 40, 60, and 80%. a** Association pathways are visualized at different population probabilities. **b** Projection pathways are visualized at different population probabilities.

## Results
### Population-based tractography of young adults
We examined the tractography of 1065 subjects in the ICBM152 space. Figure 2a shows the voxel-wise probability of the association pathways, whereas Fig. 2b shows the projection pathways. Each white matter tract is visualized by a population

probability of 20, 40, 60, and 80%, respectively. The probability was quantified by the percentage of subjects with the white matter bundle passing the ICBM152 space voxels. The abbreviations of white matter bundles are listed in Supplementary Table 1. The tractography results shown in Fig. 2 are consistent with known neuroanatomy[15] and existing tractography results[9,16,17]. The lateralization of left arcuate

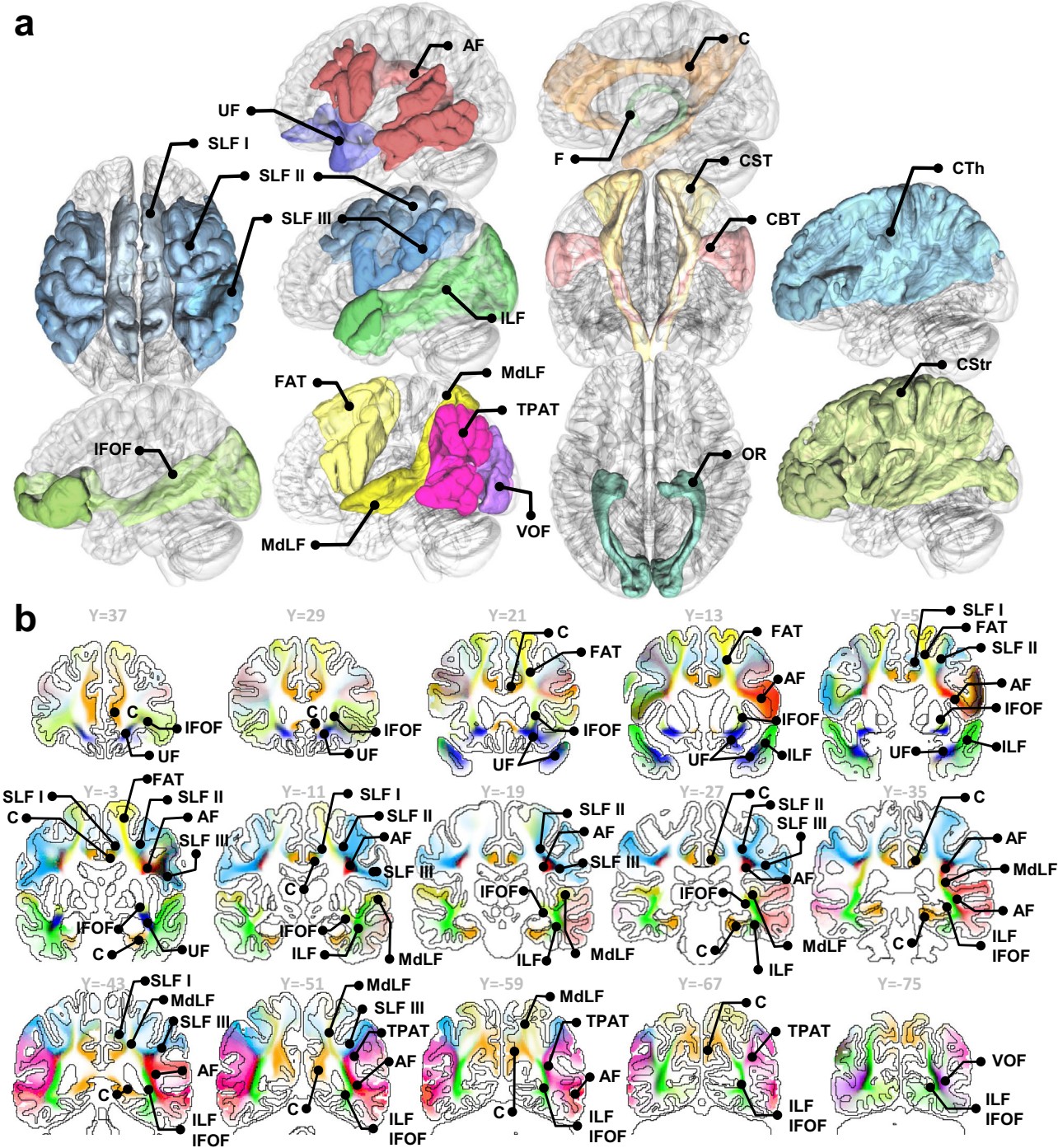

**Fig. 3 | Overview of the population-based tractography atlas in 3D rendering and slice-wise coronal sections. a** White matter pathways are visualized using 20% population probability. **b** Coronal sections of association pathways in the ICBM152 space. The color intensity scales with the population probability of the white matter bundles in the young adult population.

fasciculus (AF) can be readily observed by its substantially larger volume.

Figure 3a visualizes all white matter bundles rendered by an iso-surface of 20% population probability in the ICBM152 space. The AF and superior longitudinal fasciculus (SLF) in Fig. 3 show broader coverage than those of the projection pathways such as corticospinal tract (CST), corticobulbar tract (CBT), optic radiation (OR), and fornix (F). This result can be explained by higher between-subject differences in AF and SLF[12]. Figure 3b further shows coronal sections of tract probability. The maximum color saturation corresponds to 100% population probability, whereas white color corresponds to 0% population

probability. The probabilities of association pathways are visualized to illustrate their anatomical relation and relative location. The results are consistent with a recent population-based tractography[17].

**Tract-to-region connectome**

The tract-to-region connectome based on Brodmann areas and Kleist parcellations are shown in Fig. 4a and Fig. 4b, respectively, whereas the one based on HCP-MMP parcellations is shown in Fig. 5. The Brodmann, Kleist, and HCP-MMP parcellations have 39, 49, and 180 cortical regions. The resulting connectivity matrices have 39-by-52, 49-by-52, and 180-by-52 entries for Brodmann, Kleist, and HCP-MMP

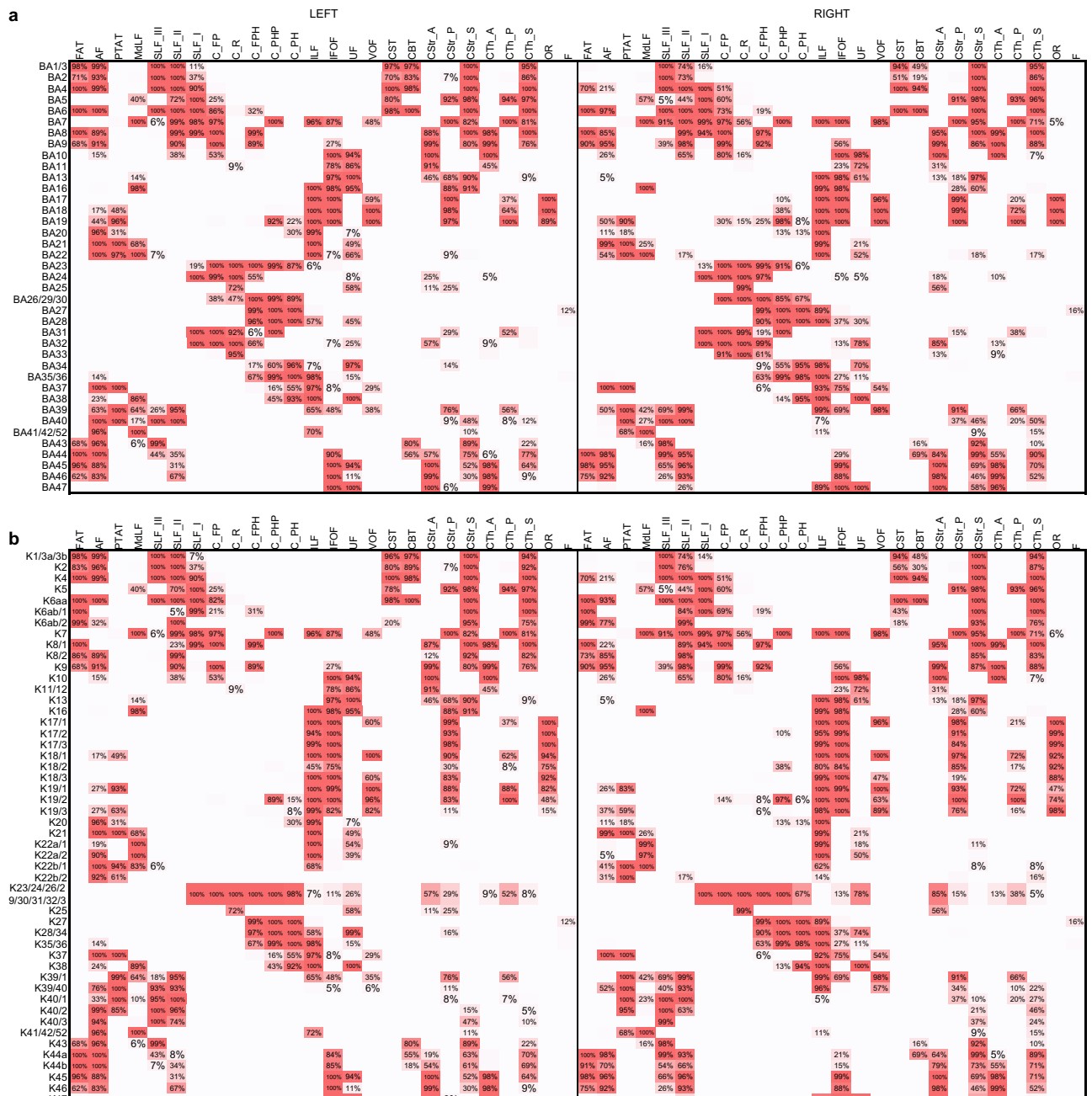

**Fig. 4 | The tract-to-region connectome matrices derived from the Brodmann and Kleist brain parcellations, respectively. a** The tract-to-region connectome using the Brodmann parcellation. **b** The tract-to-region connectome using the Kleist parcellations. The rows of the matrices correspond to each brain region defined by cortical parcellations, whereas the columns correspond to each white matter bundle. The tract-to-region connectome matrices show the population probability quantified from 1065 young adults. Probability values lower than 5% were left blank to facilitate inspection. Source data are provided as a Source Data file.

parcellations. Each row of the matrices corresponds to a cortical region, and each column corresponds to a white matter tract. The population probability was quantified by checking the corresponding cortical region and white matter tract intersection in the ICBM152 space. The left half of the matrix is the connection probabilities in the left hemisphere, whereas the right half is those in the right hemisphere. The population probabilities are color-coded by red colors: the highest saturation corresponds to the highest probability (100%), while the white color corresponds to the lowest probability (0%). The entries with less than 5% connection probability are left blank to facilitate visualization.

Most matrix entries in Brodmann (1733 out of 2028 entries, 85.45%) and Kleist parcellations (2164 out of 2548 entries, 84.93%) have population probabilities greater than 95% or smaller than 5%, meaning that these tract-to-region pairs are either connected or not connected in the majority of the study population. Around 15% of the matrix entries in Brodmann and Kleist parcellations show probabilities between 5 and 95% due to substantial individual variation. Interestingly, although HCP-MMP have more parcellation regions (180 regions) than Brodmann and Kleist parcellations (39 and 49 regions), it gives a remarkably similar figure. A total of 86.50% of its matrix entries (8096 out of 9360 entries) also have probability values greater than 95% or smaller than 5%. A

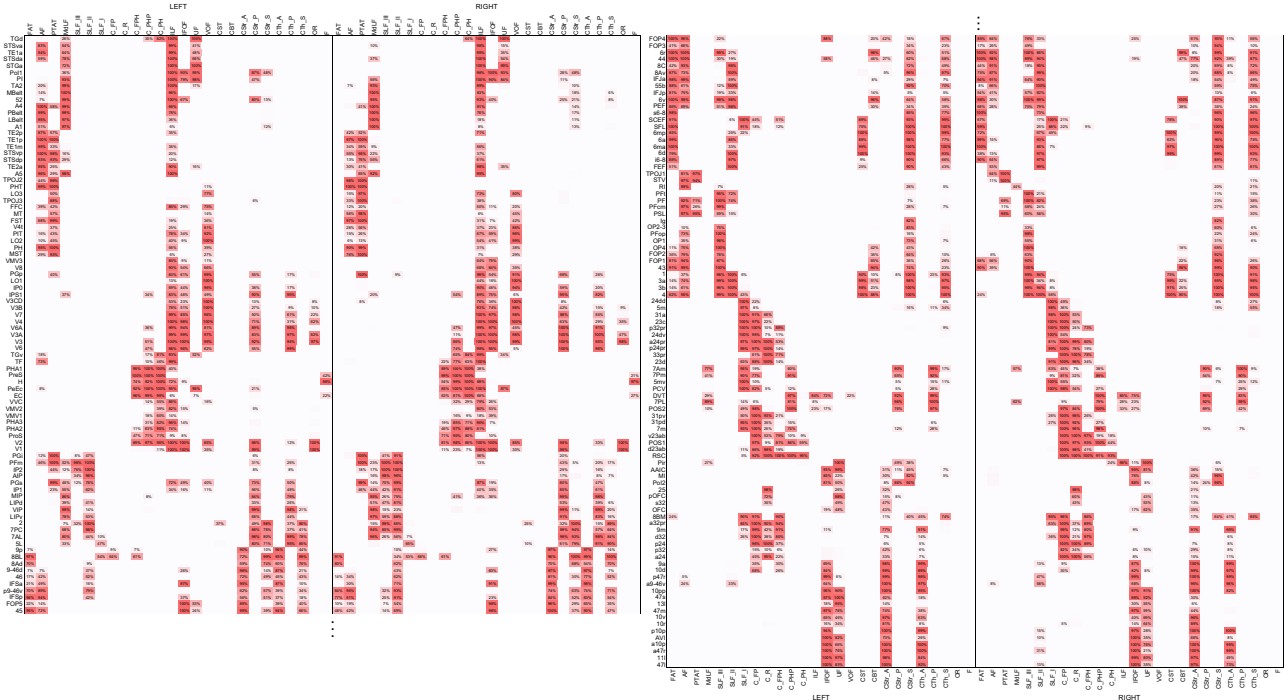

**Fig. 5 | The probabilistic tract-to-region connectome matrices derived from the HCP multimodal parcellation.** The 180 rows of the matrices correspond to each brain region defined by the HCP cortical parcellations, whereas the columns correspond to each white matter bundle. Source data are provided as a Source Data file.

random parcellation with 360 regions derived from Craddock's random parcellations[18] also showed a similar result (Supplementary Fig. 1): 87.09% of the matrix entries (8152 out of 9360) had probability values ranging between 95 and 5%. Overall, the tract-to-region connectome showed that the young adult population shares a similar connective pattern in ~85% of the tract-to-region entries. The remaining ~15% entries have substantial individual variations with population probability between 5 and 95%, thus warranting individualized mapping.

We further examined AF connections in the tract-to-region connectome using the Sankey flow diagrams shown in Fig. 6. The diagram is based on HCP-MMP, and the color saturation scales with the population probability. The connective pattern shown in Fig. 6 is consistent with the conventional view that the left AF connects Wernicke's area in the superior temporal regions (red) and Broca's area in the inferior frontal cortex (orange). Furthermore, the diagrams also show more detailed connections of left AF to the caudal dorsolateral prefrontal cortex and the inferior parietal lobule[19–23] as well as premotor/motor regions[19]. The lateralization of AF to frontoparietal (yellow), angular (green), and superior temporal regions (red) can also be seen by comparing Fig. 6a and Fig. 6b.

### Hierarchical relation of cortical regions
Figure 7 shows the similarity matrices and the derived hierarchical relations of the cortical regions defined by Brodmann (Fig. 7a) and Kleist parcellations (Fig. 7b), whereas the results for HCP-MMP are shown in Fig. 8. The column and row positions of the matrices are reordered based on clustering results to facilitate inspection. The dendrograms on the top of Figs. 7 and 8 show the hierarchical relation of the cortical regions and the vertical distance scales by the cost for merging. Overall, Figs. 7 and 8 show a consistent result, with cortical regions categorized into dorsal, ventral, and limbic networks. Although differences can be observed at each parcellation, the dorsal network includes most frontal (excluding prefrontal) and parietal regions, whereas the ventral network includes temporal and occipital regions. The limbic network comprises the prefrontal, insula, and

upper cingulum regions. In Brodmann parcellation (Fig. 7a), its dorsal network further includes the superior temporal gyrus. In contrast, the dorsal network in Kleist (Fig. 7b) and HCP-MMP (Fig. 8) only include a small posterior section of the superior temporal gyrus. The discrepancy is likely due to more detailed parcellation in Kleist and HCP-MMP at areas 22, 39, and 40. The detailed parcellation in HCP-MMP allows for revealing the subnetworks under the dorsal network, including frontal (orange colored), inferior parietal (yellowish and light green), and superior parietal (cyan) subnetworks (Supplementary Fig. 2). Similarly, HCP-MMP shows subnetworks under the ventral networks, including occipital (purple and light blue), inferior temporal (magenta), superior temporal (light red) subnetworks (Supplementary Fig. 3). The limbic networks are primarily consistent across Brodmann, Kleist, and HCP-MMP. The subnetworks cover prefrontal regions and cingulum, bridging the dorsal and ventral networks. More detailed subnetworks are shown in Supplementary Fig. 4.

### Hierarchical relation of white matter bundles
Figure 9 further shows the similarity matrix between the association pathways (Fig. 9a), and the dendrogram illustrates the hierarchical clustering results (Fig. 9b) based on the HCP-MMP tract-to-region connectome. The left and right hemispheres show highly similar hierarchical relations that group association pathways into four systems, including the arcuate system (purple), anterior ventral system (red), posterior ventral system (cyan), and cingulum system (green). The first system includes AF, SLF II, SLF III, and FAT. These pathways all connect to Broca's area and have correlated with language functions shown by several studies (detailed in the Discussion section). The second system includes MdLF, TPAT, VOF, and ILF. TPAP has several alternative naming, such as the posterior AF, posterior SLF, or SLF-TP (Supplementary Table 1). The third system includes UF and IFOF, and both are characterized by their frontal connection from the temporal and occipital lobes, respectively. The fourth system includes all cingulum pathways and SLF I, likely because the SLF I is closely adjacent to the cingulum at (Y = −3 and Y = −11) and entirely separated from SLF II

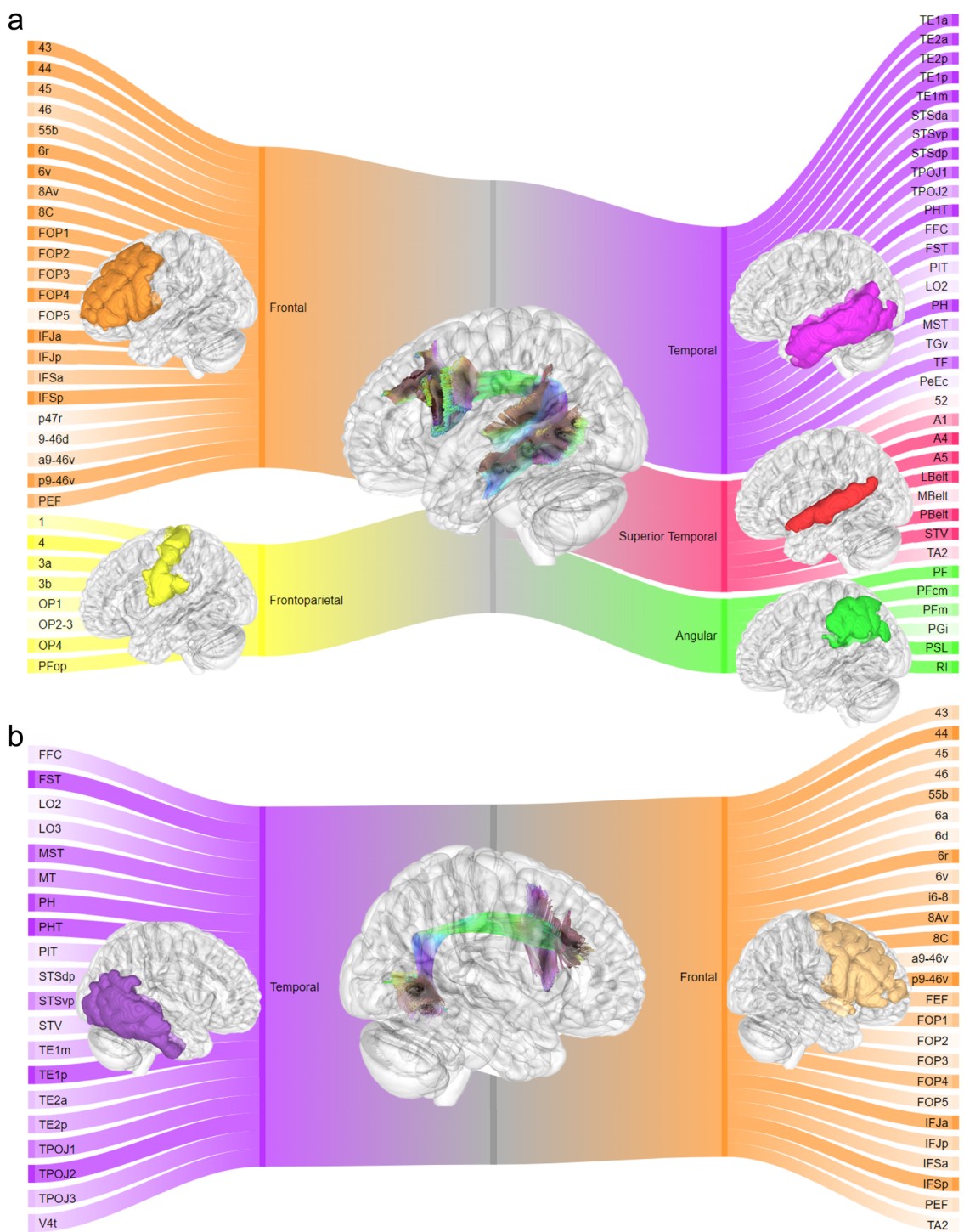

**Fig. 6 | The tract-to-region connective pattern of the arcuate fasciculus shown by Sankey flow diagrams. a** The connective pattern of the left arcuate fasciculus. **b** The connective pattern of the right arcuate fasciculus. The diagrams are based on the population probability calculated from the tract-to-region connectome in Fig. 5.

The color saturation scales with the connection probability. The left arcuate fasciculus shows substantially lateralized connections to frontal-parietal (yellow), angular (green), and superior temporal regions (red).

and III by FAT (Fig. 3b). The above data-driven clustering results showed the relation between white matter pathways based on their similarity in the tract-to-region connectome.

## Discussion

Here we quantify the tract-to-region connectome in the young adult population. The constructed matrices provide a resource for both neuroscience and clinical studies to evaluate the probability of a white

matter tract connecting to a cortical region. Based on this tract-to-region connectome, we further applied hierarchical clustering between cortical regions and between white matter association pathways to understand their relations. The results in cortical regions revealed dorsal, ventral, and limbic networks, especially using more detailed parcellations such as HCP-MMP.

The dorsal, ventral, and limbic networks can be applied to many existing functional models. The dorsal system includes most frontal

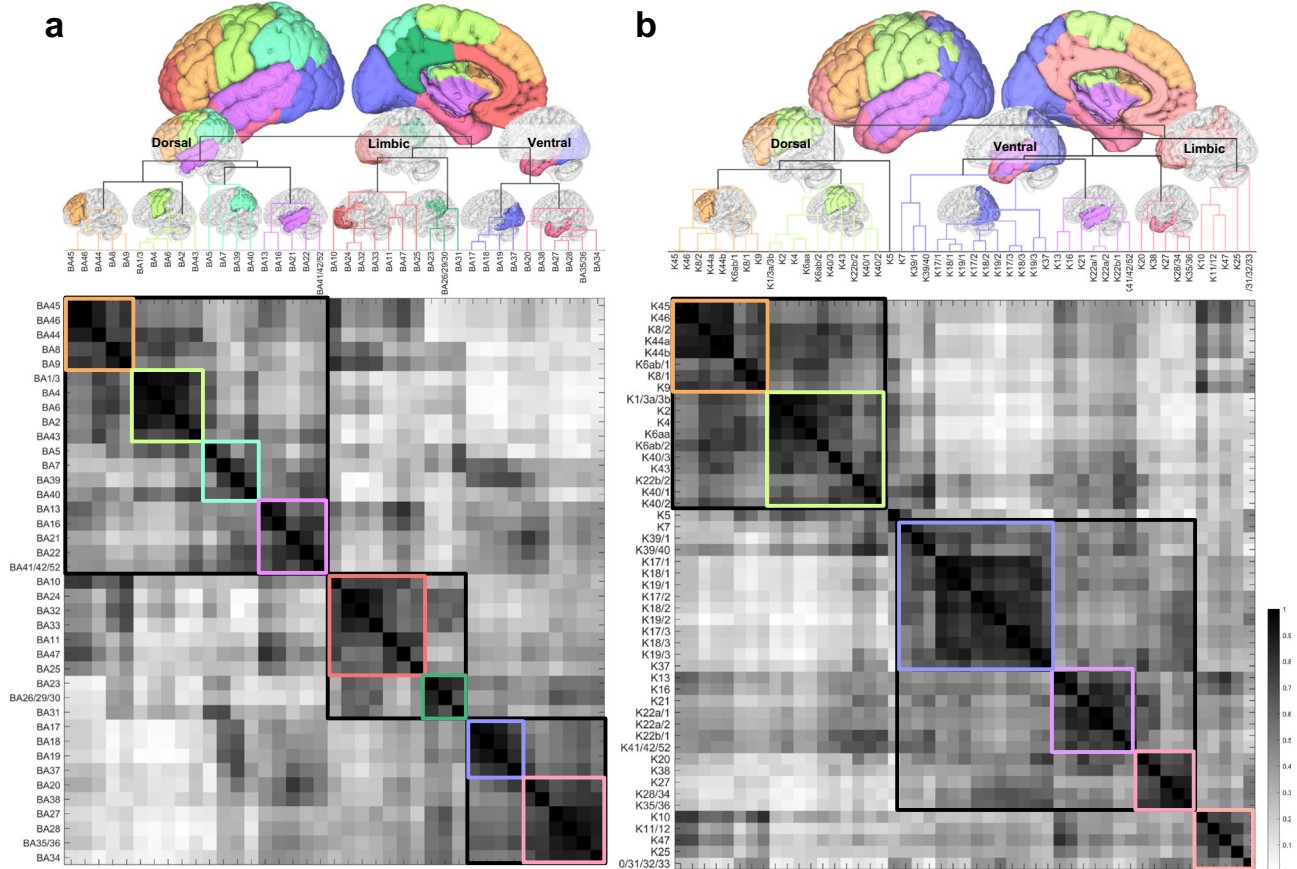

**Fig. 7 | Similarity matrices between cortical regions and derived dendrograms based on Brodmann and Kleist parcellations, respectively. a** The similarity matrix based on Brodmann parcellations. **b** The similarity matrix based on Kleist parcellations. The similarity matrices were calculated by nonparametric Spearman correlation between the row vectors of the connectome matrices. The hierarchical relation of cortical areas is then visualized using dendrograms computed by hierarchical clustering. The vertical distances in the dendrograms are scaled with the clustering cost. Both dendrograms show grossly consistent results revealing the limbic, dorsal, and ventral networks. Source data are provided as a Source Data file. MATLAB (MathWorks©, https://www.mathworks.com/) was used to create the diagram and matrix.

lobe (excluding prefrontal) and parietal lobe, whereas the ventral system includes the temporal lobe and occipital lobe. The dorsal and ventral subnetworks shared many similarities with the existing dual-stream models in the language and visual functions[24–27]. The tract-to-region connectome matrices and all intermediate data, such as tractogram of each bundle, read-to-track subject data, and source code, are publicly available at https://brain.labsolver.org. The shared data could also construct the conventional region-to-region connectome for each white matter pathway, as illustrated in our previous connectome study[28].

Neuroanatomical evidence has shown that brain regions in both human and non-human primates could be connected through more than one route[29]. Existing studies have predominately focused on region-to-region connections and simplified the role of white matter bundles as "edges" in the network model. Consequently, the region-to-region connectome falls short of illustrating the association between cortical regions and white matter pathways. This limitation becomes obvious in lesion-symptom mapping studies of aphasia: damaging Broca's area does not necessarily lead to Broca's aphasia[30,31], but lesions involving the anterior segment of the left AF are a strong symptom predictor[32]. Since cortical regions themselves may not be sufficient to explain functional deficits[33], the role of white matter pathways should be considered in dysconnectome studies[34–36]. For these studies, the tract-to-region connectome could provide a population-based reference to understand the relation between cortical regions and white matter bundles.

The tract-to-region connectome can be utilized in various scenarios in which the white matter tracts are the targets of interest. For clinical cases involving a lesion in deep white matter structures, the population probability quantified by the tract-to-region connectome can provide the likelihood of an affected white matter pathway leading to functional deficits in a cortical region. Conversely, in fMRI, EEG, or SEEG studies identifying a cortical region of interest, the tract-to-region connectome can translate the findings to their corresponding white matter pathways based on population probabilities. Subsequently, the information can delineate the circuit mechanism behind a cognitive model or verify the structure-function hypotheses. Both of them may further help explore white matter targets for neurological modulations using deep brain stimulation, focused ultrasound ablation, or laser ablation.

The tract-to-region and region-to-region connectome provide different perspectives on the organization of brain networks, as shown by their different clustering results. The clustering on the region-to-region connectome concerns whether two cortical regions are closely connected[37,38], but clustering on the tract-to-region connectome concerns whether two cortical regions share similar white matter connections. As a result, the region-to-region connectome tends to group frontal and prefrontal regions due to their strong connections through short association pathways[38,39]. In comparison, the results from the tract-to-region connectome using three differential parcellations unanimously separated prefrontal and frontal regions due to their distinctly different connections to the limbic and dorsal networks. The

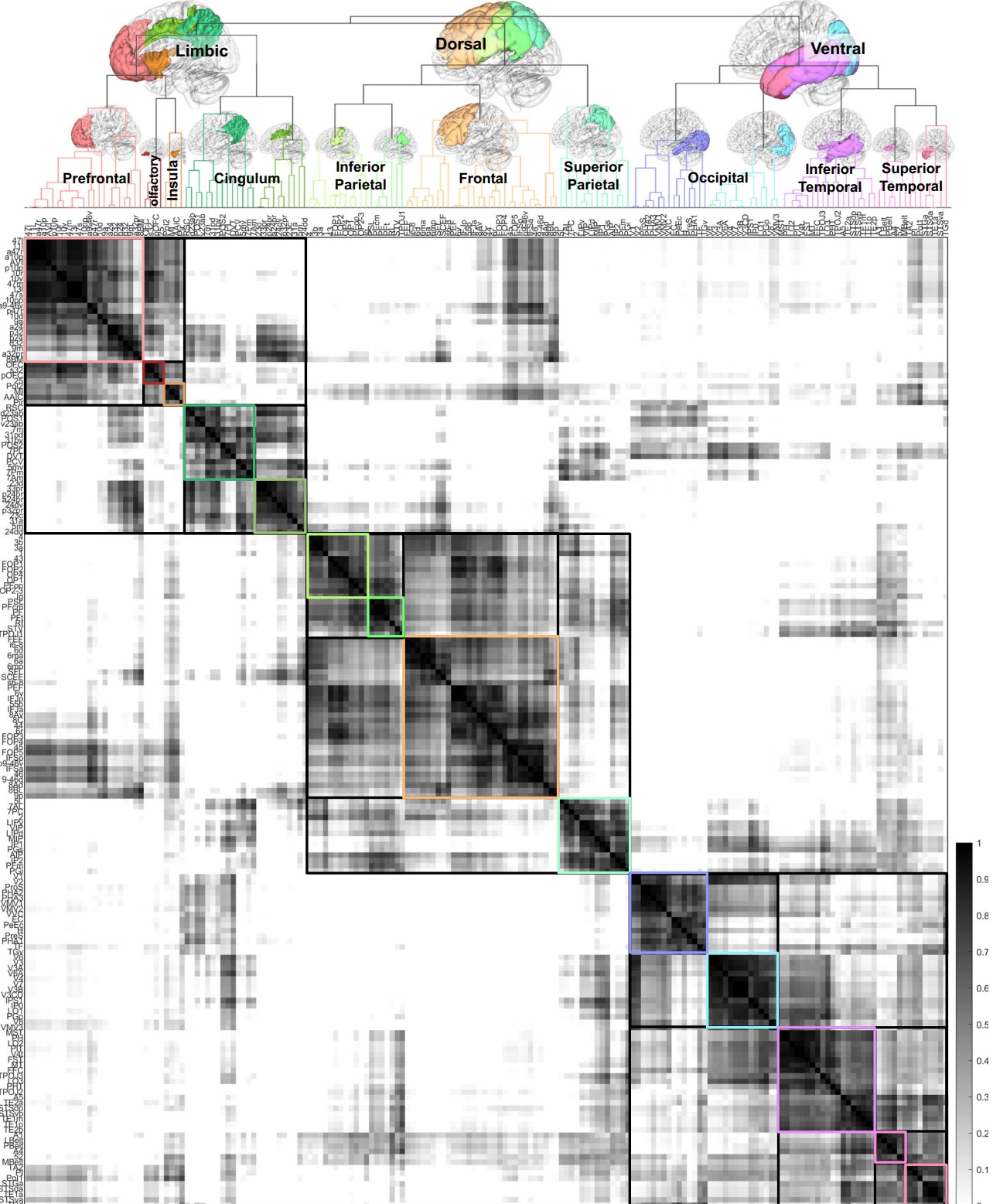

**Fig. 8 | Similarity matrix between cortical regions and its derived dendrogram based on HCP multimodal parcellation.** Consistent with the previous figure's results derived from Brodmann and Kleist parcellations, the cortical regions can be clustered into limbic, dorsal, and ventral networks. The limbic network includes the limbic system, prefrontal cortex, olfactory cortex, and insula. The dorsal network includes most of the remaining frontal lobe, parietal lobe, and part of the superior temporal gyrus, whereas the ventral network includes most of the temporal and occipital lobe. Each network has its downstream hierarchical structures of the subnetworks. Source data are provided as a Source Data file. MATLAB (Math-Works©, https://www.mathworks.com/) was used to create the diagram and matrix.

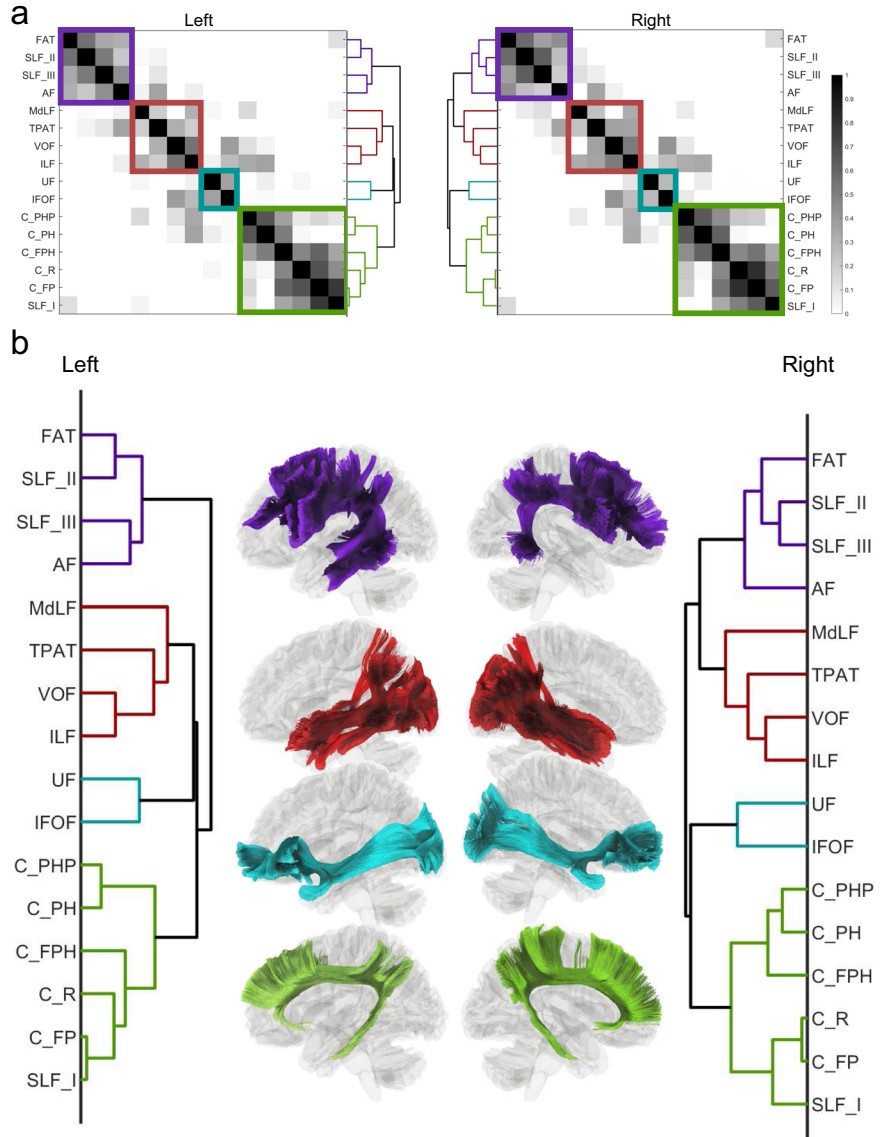

**Fig. 9 | Similarity matrices between association pathways and their derived dendrograms based on HCP multimodal parcellation. a** The similarity matrices were calculated by nonparametric Spearman correlation between the column vectors of the connectome matrix. **b** The hierarchical relation is then visualized using dendrograms computed by using hierarchical clustering. The horizontal distance in the dendrograms scales with the clustering cost. Both dendrograms show four categories of association pathways on both hemispheres, including the cingulum system (green), posterior ventral system (cyan), anterior ventral system (red), and arcuate system (purple). Source data are provided as a Source Data file. MATLAB (MathWorks©, https://www.mathworks.com/) was used to create the diagram and matrix.

prefrontal regions are closely connected with the limbic network through the cingulum, whereas frontal regions are closely connected with the dorsal network through SLF II and SLF III. The difference in clustering context will lead to entirely different results and application scenarios that answer different neuroscience questions.

We also derived the hierarchical relation between white matter bundles. While many studies have been conducted to cluster white matter tracts[40–45], these clustering methods did not consider the connective pattern with the cortical regions. The clustering in this study showed that SLF II and SLF III are closely related to AF, whereas SLF I is closely related to the cingulum. These results may appear questionable and astonishing at first glance, but there are supporting references: Catani et al.[20] showed SLF III as the anterior segment of the AF, which did not include SLF I. Wang et al.[46] suggested that the SLF I could be viewed as part of the cingulum system. From the clinical perspective, especially in the surgical intervention of brain tumors, the

neurosurgical consensus is that the eloquent area correlated with post-surgical functional deficits includes regions innervated by AF, SLF II, and SLF III[47–49]. These areas did not include SLF I because SLF I did not show significant language function[50]. Furthermore, SLF I was delineated by anterograde tract-tracer technique in rhesus macaques[51], where detailed mapping was illustrated by Schmahmann and Pandya's work[52]. In their work, among 15 cases enhancing the SLF I (cases 1, 2, 3, 4, 6, 7, 9, 17, 22, 26, 27, 28, 29, 31, and 33), 12 of them (except cases 26, 27, and 28) also enhanced the cingulum bundle as labeled by the authors. In comparison, only two cases (cases 4 and 31) enhanced SLF II, and three (cases 6, 7, and 33) enhanced SLF III. This hinted at a closer relationship between the SLF I and cingulum than with the SLF II or SLF III. Nonetheless, it is noteworthy that the clustering in this study was based on the tract-to-region connectome entries, and by no means could this be used to confirm a new naming convention or provide a new neuroanatomical definition as each bundle has well-defined

anatomical locations, as shown in Fig. 2. More functional or lesion-based studies are needed to support or refute these clustering results.

There are limitations to this study. The tract-to-region connectome did not include cerebellar, commissural, brainstem, or connections between subcortical structures. Excluding cerebellar and brainstem pathways are due to different slices coverage near the brainstem and insufficient spatial resolution to generate reliable fiber tracking results. The commissural pathways are excluded because of the limited ability of the fiber tracking methods to resolve crossing-kissing patterns when the corpus callosum crosses the hemispheres[53]. Moreover, for bundles mapped in this study, the tract-to-region relation was determined by a simple overlap in the binary mask. This setting was used to compensate for tractography's "gyral bias" that failed to map connections at the "gyral bank." However, there could be spurious connections because tractography could not confirm inner-vation. For example, the connection probabilities between IFOF and insula regions (e.g., Pol1 and 52) were just due to IFOF passing by, and further histology validation is needed. On top of these limitations, the included tracts were also subject to errors such as trajectory deviation, premature termination, and incorrect routing, as discussed in a recent review[54]. Although multiple strategies have been leveraged to reduce false connections, we cannot rule out possible errors in the current form of the tract-to-region connectome.

Furthermore, the existing graph-theoretical analysis[5] may not be readily applicable to the tract-to-region connectome because the tract-to-region concept is conceptually different from the conventional region-to-region one. While the region-to-region connectome implies an undirected graph, the tract-to-region matrix appears to be a subset of an undirected graph called a bipartite graph. The bipartite graph comprises two disjoint sets of nodes, one for tracts and one for regions. The nodes representing tracts could not be equally exchanged with nodes representing regions. Since most network measures view all nodes equally in the computation, applying these network analyses to the tract-to-region connectome could lead to questionable results. Further theoretical development is thus required to translate network measures to the tract-to-region connectome.

This study mainly focuses on the concept of the tract-to-region connectome, and the variations due to different tracking recognition tools were not investigated. The tract-to-region relation could be derived using different tools or atlases, but additional customizations may be needed to address unique technical concerns when deriving the tract-to-region relation. Specifically, tract segmentation tools often used cortical parcellations[8,55] or end regions of tracts[10] to recognize a tract. The tracts defined by cortical parcellation would show connections according to the supplied cortical parcellations, leading to circular results in the tract-to-region connectome. A solution is to use white matter trajectories to recognize tracts, but trajectories-based methods may not effectively utilize all existing atlases[9,56,57]. Most white matter atlases were voxel-based volumes that provided only masks of tract coverage and did not have trajectory coordinates needed by trajectory-based recognition. Few atlases provide trajectories coordinates for individual subjects, but a group average would be needed to minimize the individual differences. This averaging step is critical for classifiers that are sensitive to noisy data.

Nonetheless, studies have shown that tractography segmentation could be different due to anatomical views[58] or segmentation tools[59]. The differences due to tools were direct results of different input data: region-based recognitions used cortical parcellations, while trajectory-based recognition used the topology of white matter tracts. On the other hand, the differences in anatomical views were mainly due to discrepancies in the existing categorical systems and tract nomenclature[58,60]. Much of the recent disputes focus on detailed subcomponents and their categorical relations[61]. Resolving them would need new neuroanatomical evidence from tract-tracing or cadaver dissection studies. The population-based tractography and its

corresponding tract-to-region connectome are thus subject to future revisions and updates to ensure their up-to-date accuracy. Despite those discrepancies, it is noteworthy that the anatomical locations of white matter pathways are well-defined, and this study did not invent new white matter structures. The pathways mapped in this study (e.g., those shown in Figs. 2 and 3) are anatomically consistent with existing atlases from other tools and studies, and the clustering results reported were consistent across three different cortical parcellations. This consistency may support future works extending tract-to-region mapping to lifespan studies. To this end, ready-to-track data for HCP-aging, HCP-developmental, and developing HCP studies and sample processing scripts are available at https://brain.labsolver.org to assist further brain mapping endeavors.

## Methods

### Diffusion MRI acquisition
The diffusion MRI data of 1065 subjects (Fig. 1a) were acquired from the Human Connectome Project database (WashU consortium)[2]. The age range was 22–37 years, and the average age was 28.75 years. The data were acquired using a multishell diffusion scheme with three b-values at 1000, 2000, and 3000 s/mm² with 90 sampling directions for each shell. The spatial resolution was 1.25 mm isotropic. The detailed acquisition parameters are listed in the consortium paper[2]. The preprocessed data were used. The gradient nonlinearity was corrected for each diffusion-weighted signal at each voxel by $S' = b_0 \left( \frac{S}{b_0} \right)^{(1/\|\mathbf{G} \cdot \mathbf{b}\|^2)}$,[62] where $S$ is the raw signal, and $S'$ is the corrected diffusion signal. $b_0$ is the b0 signal, and $\mathbf{b}$ the diffusion gradient direction. $\mathbf{G}$ is the 3-by-3 gradient nonlinearity matrix after adding an identity matrix. This per signal correction allows for keeping the original shell structures of the b-table to enable shell-based diffusion modeling.

### Diffusion MRI reconstruction
The diffusion data were linearly rotated to align with the ac-pc line of the ICBM152 space and simultaneously interpolated at 1 mm using cubic spline interpolation. The b-table was also rotated accordingly. The rotated data were then reconstructed using generalized q-sampling imaging[63] with a diffusion sampling length ratio of 1.7. An automatic quality control routine was adopted to check the b-table orientation and ensure its accuracy[64]. The reconstruction results (Fig. 1b) were further used in automated tractography.

### Automated tractography
For each subject, 52 white matter bundles were mapped using the automated tractography pipeline in DSI Studio (http://dsi-studio.labsolver.org, developed by the author), which combined deterministic fiber tracking algorithm[65], randomized parameter saturation[12], topology-informed pruning[66], and trajectory-based tract recognition[12] (detailed in the next section) as an integrated interface. The default settings were used: the anisotropy threshold was uniformly and randomly selected from 0.5 to 0.7 Otsu threshold. The angular threshold was uniformly and randomly selected from 15 to 90 degrees. The step size was uniformly and randomly selected from 0.5 to 1.5 voxel spacing. The minimum length was 30 mm. The tracking was repeated until the ratio of streamlines to voxels reached 1.0.

### Trajectory-based tract recognition
The tract recognition used the nearest neighbor method to classify tracts. Since the nearest neighbor classifier is sensitive to noisy data and prone to overfitting, training data preparation was critical for best performance. Most population-based atlases have substantial individual variations and thus would require additional averaging to minimize individual differences. Therefore, in this study, the recognition used a population-averaged tractography atlas[28], which was aggregated from the young adult population and vetted by a team of

neuroanatomists. An updated version of the atlas in the ICBM152 nonlinear asymmetry space (publicly available at https://brain.labsolver.org) was used in this study.

For each subject, the tractography atlas was nonlinearly warped to subject space using the diffeomorphic mapping derived between the subject's anisotropy image and the ICBM152-space anisotropy image. The Hausdorff distance[67] was computed between each subject and the atlas tract. The shortest distance then determined the label of subject tracts. Some tracts might substantially deviate from all atlas trajectories, and thus a maximum allowed Hausdorff distance (termed tolerance distance) of 16 was used to remove them. The minimum Hausdorff distance was increased to 18 and 20 mm if no bundle was found after topology-informed pruning[66]. In this study, 4 out of the 52 bundles did not reach a 100% yield rate for all 1065 subjects: left CBTs (983/1065), right CBTs (1054/1065), right AF (1046/1065), right occipital corticopontine tract (1064/1065). The missing of the corticobulbar and corticopontine tracts in some subjects could be due to the limitation of the fiber tracking algorithm to capture substantial turning of the pathways. The right AF in some subjects was entirely labeled as SLF III due to no connection to the superior temporal lobe.

The steps mentioned above, including subject-space fiber tracking, parameter saturation, randomized parameters, topology-informed pruning, anisotropy-based warping, Hausdorff distance computation, were integrated as the automated tractography function in DSI Studio. The source code is also available on GitHub repository at https://github.com/frankyeh/DSI-Studio.

The computation was conducted at the Pittsburgh Supercomputing Center provided through the Extreme Science and Engineering Discovery Environment (XSEDE) resource[68]. The tractography result for each subject and each white matter tract are shared on http://brain.labsolver.org.

### Brain parcellations and tract-to-region connectome

The white matter bundles of each subject were then exported to the ICBM152 2009 nonlinear space to facilitate integration across the entire subject group. As shown in Fig. 1d, for each of the 1065 subjects, the trajectories of white matter bundles in the ICBM152 space were examined with the Brodmann, Kleist, and HCP-MMP parcellations to derive the population-based tract-to-region connectome. A random parcellation was also used as a comparison.

We used the ICBM152-space version of newly reconstructed Brodmann and Kleist atlases[13]. On the other hand, the ICBM152-space version of HCP-MMP was obtained from https://neurovault.org/collections/1549/ (asymmetric, improved reconstruction) and further inspected for each cortical region to manually remove the cross-sulci leakage using DSI Studio. The revised version of the HCP-MMP was shared with the DSI Studio package and is publicly available at http://dsi-studio.labsolver.org. The random parcellation was derived from level 33 of Craddock's fine-grained random parcellations[18] by assigning the labels to a gray matter mask in the ICBM152 space using the shortest distance.

A binary tract-to-region connection matrix was obtained for each subject by calculating the intersection between the voxel-wise mapping of white matter bundles and cortical regions (Fig. 1e). The binary matrices of 1065 subjects were then aggregated to compute the population probability of the tract-to-region connection, and one matrix was generated for each parcellation, respectively. The tract-to-region connectome can be downloaded from http://brain.labsolver.org.

### Hierarchical clustering

We used row vectors of the tract-to-region matrices as the feature vectors to derive the hierarchical relation between cortical regions (Fig. 1f). The similarity matrices between each region pair were quantified by their correlation. Since the correlation could be nonlinear, we used the nonparametric Spearman's rank correlation to consider possible nonlinearity relations in population probability. The hierarchical clustering was conducted using weighted average distance[69] provided by the "linkage" function in MATLAB to avoid the high variability drawback of simple single linkage clustering. For each cortical parcellation (Brodmann, Kleist, HCP-MMP), a dendrogram was generated to reveal the hierarchical relation of the cortical regions. The hierarchical clustering for white matter bundles was conducted using the column vector of the HCP-MMP connectome matrices (Fig. 1g). The clustering routine also used weighted average distance to generate the dendrogram to reveal the hierarchical relation of white matter bundles.

### Reporting summary

Further information on research design is available in the Nature Research Reporting Summary linked to this article.

## Data availability

The tract-to-region matrices, similarity matrices, clustering code, and scripts to generate identical figures are provided in the "Source Data.zip". The population-based tractography and tract-to-region connectome are publicly available at http://brain.labsolver.org. Source Data are provided with this paper.

## Code availability

The analysis tool DSI Studio[70] and its source code are available at https://github.com/frankyeh/DSI-Studio.

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

## Acknowledgements

This work used the Extreme Science and Engineering Discovery Environment (XSEDE), which is supported by the National Science Foundation grant number ACI-1548562. The computation resources were allocated under TG-CIS200026. The author was supported by NIH grant R01 NS120954. Data were provided in part by the Human Connectome Project, WU-Minn Consortium (Principal Investigators: David Van Essen and Kamil Ugurbil; 1U54MH091657) funded by the 16 NIH Institutes and Centers that support the NIH Blueprint for Neuroscience Research; and by the McDonnell Center for Systems Neuroscience at Washington University.

## Author contributions

F.-C.Y. did the data analysis and wrote the manuscript.

## Competing interests

The author declares no competing interests.

## Additional information

**Supplementary information** The online version contains

supplementary material available at https://doi.org/10.1038/s41467-022-32595-4.

