## [Peer Review File · Nature Communications]

Population-Based Tract-To-Region Connectome of the Human Brain and Its Hierarchical TopologyREVIEWER COMMENTS

Reviewer #1 (Remarks to the Author):

This is a very interesting and timely paper. It presents a simple but persuasive (and novel) idea regarding the potential of studying the connectome according to the white matter tracts and their connections to cortical regions at the population level. The results are quite attractive and make an important point that network-based connectome analyses ignore the information about anatomically named white matter tracts. Instead of essentially counting streamlines between cortical regions (as in traditional connectome methods), this new connectome records the presence of tracts that intersect each region. Ideas are presented for visualization of tracts and cortical regions based on hierarchical clustering using the information in this new style of connectome. I especially like Figure 6 that gives a visualization of the arcuate connectivity pattern in each hemisphere. This is a cool and impactful work deserving of publication. My (mostly minor) comments follow.

Line 56 (Introduction) “Irrelevant connections” does not have a clear meaning. The reader wonders if these are false positive connections as determined by experts, or non-reproducible connections in test-retest, or spurious fibers that are outliers on the population level somehow, or if they are something else. Perhaps given the rest of the sentence, the author means “non-reproducible connections?” This should be further clarified in the methods.

The word “atlas” is used a lot and gets a bit confusing. Fig 3 talks about a “population-based tractography atlas” but this 1k subjects tract data in Fig 3 was created by applying another atlas. Is this 1k subjects data released as an atlas? Then later there are several cortical atlases. This leads to a bit of confusion at times when the word atlas is used without saying what atlas.

Figure 9 caption “The hierarchical relation of cortical areas is then visualized..” I think this is a typo. This figure visualizes a hierarchical relation of some sort between association fiber tracts.

Line 208 typo “shown in this stud”

It should be discussed or mentioned somewhere that this connectome analysis is restricted to 52 tracts and presumably all other tractography streamlines are removed. This likely removes (for example) cerebellar connections, superficial connections, and streamlines that are debatably part of tracts like corticospinal (which may or may not connect to cortex outside of primary motor depending on which paper is read...) It is non-trivial to define the tracts and of course it has been done well here, but some discussion is warranted about this and about what was removed.

It should probably be discussed that a different definition of tracts, a different algorithm for detecting them, etc. would produce different results. This manuscript seems to take the perspective that the tract identification results are very correct. Also, no results are reported about the success of identifying the tracts. Were all tracts identified in all subjects?

Minor typos in singular/plural and occasional absence of “the” should be checked throughout the manuscript.

“The preprocessed data were used and further corrected for gradient
237 nonlinearity.” How? Using what?

The clustering methods are light on rationale for why the choices were made. For example why the nonparametric Spearman’s rank correlation? Is this accepted in another connectome analysis or considered best for this particular type of data for some reason? Obviously all these choices will affect the results.

Line 422 “Fifugres” typo

The paper might benefit from the addition of a final sentence or conclusion to help the reader appreciate the overall impact.

Reviewer #2 (Remarks to the Author):

The author presents a new approach for investigating cortical connectivity, not by analyzing which cortical regions are connected (region to region connectivity), which is the typical approach known from literature, but rather by analyzing which white matter fiber tracts are connected to which cortical region (tract to region connectivity). The rationale behind this is that classic connectomics (region to region connectivity analysis) ignores how individual white matter tracts contribute to the connections of the cortical regions and thereby discards information that might be important for many research questions. The tract to region connectome is presented as a method that provides information complementary to classic structural and functional connectivity.

The approach is new and seems very promising. The author shows that the generated information is complementary to existing connectivity analysis techniques.

The author also roughly describes exemplary use cases for the new approach, such as the analysis of the neuroanatomic relation of tract components (here SLF I-III) to other tracts or tract systems (here the cingulum and the arcuate fasciculus), or the analysis of the sources of aphasia. Unfortunately, for the latter example it remains unclear how the presented tract to region connectivity will provide additional information compared to classical region to region connectivity, particularly since the tract to region connectivity as it is presented here is binary at the subject level. This means that it can only show that a specific tract is affected by a certain pathology if the pathology causes a complete disconnection of a region and the respective tract. Maybe the author can comment on this. In general it would be desirable to have a more detailed description of scenarios where the presented concept could provide valuable information to fully judge the significance of the method for the research field.

The methodology of the work seems sound, but is in parts described very superficially. Particularly the automated tractography is not reproducible with the information provided in the manuscript.

While the conducted experiments are sound, several questions arose while studying the manuscript that are not covered by the experiments and also not discussed:

- The author mentions that around 15% of the observed tract to region connections occur often but not always (likelihood between 5% and 95%) and attribute this to “substantial individual variation”. This is the case for all used cortical parcellations. This tells me that the cause might not be “individual variation” but maybe simply caused by some factor in the tractography pipeline, such as noise and an insufficient resolution, that causes tracts that actually only connect to N regions to randomly “leak” to neighboring regions. Whatever the reason may be, it would be interesting to look into this further. One way to gain more information about this, apart from actually looking at these cases, could be the use of random but parameterizable parcellations instead of actual atlas-based parcellations.

- In the approach presented by the authors, the information which brain region is connected to which other region is lost. Instead of completely skipping this information, it would have been straightforward to extend classic region to region connectivity with the tract specific information included in the presented tract to region connectivity, e.g. representable by a 3D connectivity matrix (“region to region via tract connectivity”). Why did the author choose to discard this information?

- The author only mentions the tract atlas used for the automated tractography does not describe it in detail. Could the choice of this atlas affect the results? Are there alternatives? Why was this atlas chosen? This should be discussed.

- The author describes that a tract and a region are regarded as connected as soon as their binary masks overlap. In this scenario it seems possible that regions connect to parts of the tracts that are actually not endpoints. This should be discussed.

Overall the work is well presented and the approach seems very interesting for the research community and I recommend acceptance of the manuscript if the above issues and comments are addressed.

Reviewer #3 (Remarks to the Author):

In this paper the author present a new way to present and analyse structural connectome data of the human brain. The paper is well written and figures, in particular, are very clear and easy to understand. However, I'm not sure I fully understand what is the main outcome of this study.

Segmenting white matter tract using automatic and hierarchical clusters methods has been presented in several previous works. The novelty here is that a connectome-like analysis is applied to a predefined set of tracts extracted using an automatic approach. I fully agree with the author that connectomes built using only a region-to-region approach may not provide enough information of the real underlying white matter connectivity and may lead to a "black-box" analysis approaches. However, the usual definition of connectome also implies that we are also trying to look at the entirety of white matter connections (within data resolution limits, of course). However, by selecting only a subset of white matter tracts as provided by the automatic segmenting tool I don't think we are getting an unbiased connectome view and everything will be highly depended on how these tracts are extracted. Moreover, all classical network properties will be biased by the actual number and selections of tracts. Change the set of tracts or how they are extracted and the topology will likely be different.

Not much information is given about how these tracts are selected and extracted and I think this the main factor driving all the results in the manuscript. The author should significantly expand this part. Per se, the tract-to-region approach is more a way to very nicely display how presegmented tracts connect different brain regions. A big concern I have here is that depending on the segmenting algorithm most of the tracts may be already defined, in first place, by cortical regions making the analysis presented in this manuscript a circular reasoning problem. The constant sharp 95%/5% of connectivity ratio for some of the tract-to-region makes me suspect this can be the case.

The use of different cortical parcellations only in part can solve this problem because of the significant overlap of many of the major cortical regions. A more appropriate analysis would have been to compare different segmentation tools and show if and how the connectome topology modify itself. If the tract-to-region approach is providing meaningful results, a consistent pattern should emerge.

Looking at the clustering results presented here I'm surprised to read that the SLFI is part of the Cingulum system. The Cingulum is very well anatomically defined and it is running within the cingulate gyrus or surrounded by the cingulate cortex. On the contrary SLF1 is more lateral-superior and connects superior frontal lobe with mostly the superior parietal lobule and other parietal regions. Again, depending on how the tracts are defined/extracted here, some of the termination may be shared but not all tracts have only two clear termination points. Some have multiple branches and some like the cingulum may multiple U-shape fibres running within the tract itself. Similarly, while SLF3 can be associated to the arcuate system I'm not sure SLF2 can be said to belong to anatomically or functionally to the arcuate system. I can't find justification of that in the references provided. Also I don't think that merging together IFOF with Uncinate as a system, just because they share the same frontal region, is a way to improve our understanding on human brain anatomy.

I think that while some of these anatomical claims are interesting, some are also misleading and driven by how the specific set of tracts have been dissected. I think that strong anatomical claims should not just coming from a pure data-driven approach but they need to be strongly supported and validated using independent anatomical or clinical data.

Reviewer #1:

General Comment:

This is a very interesting and timely paper. It presents a simple but persuasive (and novel) idea regarding the potential of studying the connectome according to the white matter tracts and their connections to cortical regions at the population level. The results are quite attractive and make an important point that network-based connectome analyses ignore the information about anatomically named white matter tracts. Instead of essentially counting streamlines between cortical regions (as in traditional connectome methods), this new connectome records the presence of tracts that intersect each region. Ideas are presented for visualization of tracts and cortical regions based on hierarchical clustering using the information in this new style of connectome. I especially like Figure 6 that gives a visualization of the arcuate connectivity pattern in each hemisphere. This is a cool and impactful work deserving of publication. My (mostly minor) comments follow.

Responses: Thank you for the encouraging comments and suggestions. The manuscript has been substantially revised. The following are point-to-point responses.

Comment 1:

Line 56 (Introduction) "Irrelevant connections" does not have a clear meaning. The reader wonders if these are false positive connections as determined by experts, or non-reproducible connections in test-retest, or spurious fibers that are outliers on the population level somehow, or if they are something else. Perhaps given the rest of the sentence, the author means "non-reproducible connections?" This should be further clarified in the methods.

Responses:

The sentence was revised to correct this confusing term. To summarize here: if a tract has a large Hausdorff distance away from the expert-vetted tractography atlas, then the tracts were regarded as "irrelevant" and discarded. This approach has been shown to produce high reproducibility results in another study (Yeh 2021).

Modifications Made:

[Instruction Section]

Several technical advances were used to construct the tract-to-region connectome (Fig. 1). The white matter bundles of the 1065 young adults were mapped using automated tractography. Although many automated tractography methods are available⁶⁻¹¹, most have used cortical parcellation to recognize tracts. These *region-based* methods could lead to circular analysis in the tract-to-region connectome. Therefore, this study used *trajectory-based* recognition¹² and did not filter tractogram by brain regions. After trajectory-based recognition, connections substantially deviated from the expert-vetted tractography atlas were removed.

Comment 2:

The word "atlas" is used a lot and gets a bit confusing. Fig 3 talks about a "population-based tractography atlas" but this 1k subjects tract data in Fig 3 was created by applying another atlas. Is this 1k subjects data released as an atlas? Then later there are several cortical atlases. This leads to a bit of confusion at times when the word atlas is used without saying what atlas.

Responses:

The manuscript was revised accordingly.

The term *cortical atlas* was replaced with *parcellations*, whereas *atlas* was only used to describe the expert-vetted tractography atlas (Yeh 2018) used in tract recognition.

The 1065 subjects' data are released as the results, but they were not called "atlas" in this study.

Modifications Made:

The modifications could not be listed here due to widespread corrections throughout the manuscript.

Comment 3:

Figure 9 caption "The hierarchical relation of cortical areas is then visualized.." I think this is a typo. This figure visualizes a hierarchical relation of some sort between association fiber tracts.

Response: revised accordingly.

Comment 4:

Line 208 typo "shown in this stud"

Response: revised accordingly.

Comment 5:

It should be discussed or mentioned somewhere that this connectome analysis is restricted to 52 tracts and presumably all other tractography streamlines are removed. This likely removes (for example) cerebellar connections, superficial connections, and streamlines that are debatably part of tracts like corticospinal (which may or may not connect to cortex outside of primary motor depending on which paper is read...) It is non-trivial to define the tracts and of course it has been done well here, but some discussion is warranted about this and about what was removed.

Responses:

Thank you for the suggestion.

The Discussion section is revised to provide a detailed discussion about the challenge and limitation accordingly.

Modifications Made:

[Discussion Section]

There are limitations in this study. The tract-to-region connectome did not include cerebellar, commissural, brainstem, or connections between subcortical structures. Excluding cerebellar and brainstem pathways are due to different slices coverage near the brainstem and insufficient spatial resolution to generate reliable fiber tracking results. The commissural pathways are excluded because of the limited ability of the fiber tracking methods to resolve crossing-kissing patterns when the corpus callosum crosses the hemispheres⁵³. Moreover, for bundles mapped in this study, the tract-to-region relation was determined by a simple overlap in the binary mask, and there could be false-positive results. On top of these limitations, the included tracts were also subject to errors such as trajectory deviation, premature termination, incorrect routing, as discussed in a recent review⁵⁴. Although multiple strategies have been leveraged to reduce false connections, we cannot rule out possible errors in the current form of the tract-to-region connectome.

Comment 6:

It should probably be discussed that a different definition of tracts, a different algorithm for detecting them, etc. would produce different results. This manuscript seems to take the perspective that the tract identification results are very correct. Also, no results are reported about the success of identifying the tracts. Were all tracts identified in all subjects?

Responses:

This is a critical point, as the results would be different if we had used a different atlas or algorithm.

The manuscript is substantially revised to avoid assuming that the results are without possible errors. More discussions on limitations have been added to the Discussion section.

The missing information about the success rate of identifying the tracts is also added. To summarize here, most pathways have 100% identifications rate, whereas 4 pathways did not achieve 100%. The possible causes were explored and discussed.

Modifications Made:

[Discussion Section]

Because of several practical considerations, we could not comprehensively explore how a different tractography atlas or a segmentation tool affects the tract-to-region connectome. Most tract segmentation tools used *region-based* recognition. They relied on cortical parcellations^{8,55} or end regions of tracts¹⁰ to recognize a tract. This approach would interfere with the tract-to-region interpretation and lead to circular findings in the tract-to-region connectome. This study customized a *trajectory-based* recognition method to recognize each trajectory without resorting to tract clustering, which could also override the clustering analysis applied to the connectome. In addition to limitations in tools, existing atlases^{56,57} were either *voxel-based* or annotated in individuals, as summarized in a recent study⁹. The *voxel-based* atlases provided only masks of tract coverage and did not have trajectory data needed by trajectory-based recognition. Constructing the tract-to-region connectome required a *tract-based* atlas aggregated from a population in the ICBM152-space to minimize individual differences or bias²⁸, and not all atlases are fit for this purpose. The one used in this study was scrutinized to satisfy these requirements.

Nonetheless, studies have shown that the tractography segmentation could be different due to anatomical views⁵⁸ or segmentation tools⁵⁹. The differences due to tools were direct results of different input data: *region-based* recognitions used cortical parcellations, while *trajectory-based* recognition used the topology of white matter tracts. On the other hand, the differences in anatomical views were mainly due to discrepancies in the existing categorical systems and tract nomenclature^{58,60}. While there is no ambiguity in the anatomical locations of white matter pathways, much of the recent disputes focus on detailed subcomponents and their categorical relations⁶¹. Resolving them would need new neuroanatomical evidence from tract-tracing or cadaver dissection studies. The population-based tractography and its corresponding tract-to-region connectome are thus subject to future revisions and updates to ensure their up-to-date accuracy.

[Methods Section]

In this study, 4 out of the 52 bundles did not reach 100% yield rate for all 1065 subjects: left corticobulbar tracts (983/1065), right corticobulbar tract (1054/1065), right arcuate fasciculus (1046/1065), right occipital corticopontine tract (1064/1065). The missing of the corticobulbar and corticopontine tracts in some subjects could be due to the limitation of the fiber tracking algorithm to capture substantial turning of the pathways. The right arcuate fasciculus in some subjects was entirely labeled as SLF III due to no connection to the superior temporal lobe.

Comment 7:

Minor typos in singular/plural and occasional absence of “the” should be checked throughout the manuscript.

Response: The manuscript was examined to correct them.

Comment 8:

“The preprocessed data were used and further corrected for gradient 237 nonlinearity.” How? Using what?

Responses:

The Methods section was revised to include the details. To summarize, the per-voxel jacobian matrix from the HCP data were used to correct the diffusion signals at for each voxel.

Modifications made:

[Methods Section]

The gradient nonlinearity was corrected for each diffusion-weighted signal at each voxel by $S' = b_0 \left(\frac{S}{b_0}\right) \left(1/\|G\hat{b}\|^2\right)$,⁵⁹ where S is the raw signal, and S' is the corrected diffusion signal. b_0 is the b_0 signal, and \hat{b} the b vector. G is the 3-by-3 gradient nonlinearity matrix after adding an identity matrix. This per signal correction allows for keeping the original shell structures of the b -table to enable shell-based diffusion modeling.

Comment 9:

The clustering methods are light on rationale for why the choices were made. For example why the nonparametric Spearman's rank correlation? Is this accepted in another connectome analysis or considered best for this particular type of data for some reason? Obviously all these choices will affect the results.

Responses:

Spearman's rank correlation is widely used in correlation analysis to address the nonlinearity problem. Google Scholar search results using "Spearman's rank+connectome" shows 2380 findings. Since the correlations between the tract-to-region matrix entries are likely to be nonlinear, the conventional linear regression-based correlation may be questioned for violating the linearity assumption.

This rationale is included in the Methods section to strengthen the thoughts behind the experiment design.

Modifications Made:

[Methods Section]

The similarity matrices between each region pair were quantified by their correlation. Since the correlation could be nonlinear, we used the nonparametric Spearman's rank correlation to consider possible nonlinearity relations in population probability. The hierarchical clustering was conducted using weighted average distance 67 provided by the linkage function in MATLAB to avoid the high variability drawback of simple single linkage clustering

Comment 10:

Line 422 "Fifugres" typo

Responses:

Corrected accordingly.

Comment 11:

The paper might benefit from the addition of a final sentence or conclusion to help the reader appreciate the overall impact.

Responses:

Thank you for the suggestion. One paragraph was added to summarize the overall impact, and a conclusion paragraph was added to mention potential improvements for future studies.

Modifications Made:

[Discussion Section]

The tract-to-region connectome can be utilized in various scenarios in which the white matter tracts are the targets of interest. For clinical cases involving a lesion in deep white matter structures, the population probability quantified by the tract-to-region connectome can provide the likelihood of an affected white matter pathway leading to functional deficits of a cortical region. Conversely, in fMRI, EEG, or SEEG studies identifying a cortical region of interest, the tract-to-region connectome can translate the findings to their corresponding white matter pathways based on population probabilities. Subsequently, the information can delineate the circuit mechanism behind a cognitive model or verify the structure-function hypotheses. Both of them may further help explore new white matter targets for neurological modulations using deep brain stimulation, focused ultrasound ablation, or laser ablation.

[Discussion Section]

Nonetheless, studies have shown that the tractography segmentation could be different due to anatomical views⁵⁸ or segmentation tools⁵⁹. The differences due to tools were direct results of different input data: *region-based* recognitions used cortical parcellations, while *trajectory-based* recognition used the topology of white matter tracts. On the other hand, the differences in anatomical views were mainly due to discrepancies in the existing categorical systems and tract nomenclature^{58,60}. While there is no ambiguity in the anatomical locations of white matter pathways, much of the recent disputes focus on detailed subcomponents and their categorical relations⁶¹. Resolving them would need new neuroanatomical evidence from tract-tracing or cadaver dissection studies. The population-based tractography and its corresponding tract-to-region connectome are thus subject to future revisions and updates to ensure their up-to-date accuracy.

Reviewer #2:

General Comment:

The author presents a new approach for investigating cortical connectivity, not by analyzing which cortical regions are connected (region to region connectivity), which is the typical approach known from literature, but rather by analyzing which white matter fiber tracts are connected to which cortical region (tract to region connectivity). The rationale behind this is that classic connectomics (region to region connectivity analysis) ignores how individual white matter tracts contribute to the connections of the cortical regions and thereby discards information that might be important for many research questions. The tract to region connectome is presented as a method that provides information complementary to classic structural and functional connectivity.

The approach is new and seems very promising. The author shows that the generated information is complementary to existing connectivity analysis techniques.

Response:

Thank you for the encouraging comments and insightful suggestions to help improve the study substantially. Additional analysis has been conducted according to the suggestions and included in the manuscript. The following is the point-to-point responses:

Comment 1:

The author also roughly describes exemplary use cases for the new approach, such as the analysis of the neuroanatomic relation of tract components (here SLF I-III) to other tracts or tract systems (here the cingulum and the arcuate fasciculus), or the analysis of the sources of aphasia. Unfortunately, for the latter example it remains unclear how the presented tract to region connectivity will provide additional information compared to classical region to region connectivity, particularly since the tract to region connectivity as it is presented here is binary at the subject level. This means that it can only show that a specific tract is affected by a certain pathology if the pathology causes a complete disconnection of a region and the respective tract. Maybe the author can comment on this. In general it would be desirable to have a more detailed description of scenarios where the presented concept could provide valuable information to fully judge the significance of the method for the research field.

Responses:

Thank you for the suggestion to broaden the impact of this study, and paragraphs have been added to the Discussion providing examples for possible applications.

To summarize here, the critical difference is the *white matter tract*. The tract-to-region connectome could provide critical information when white matter tracts are the targets of interest. The scenarios include structure-function modeling involving the white matter pathways and clinical neurological modulations such as DBS, laser or focused ultrasound ablations.

Modifications Made:

[Discussion Section]

The tract-to-region connectome can be utilized in various scenarios in which the white matter tracts are the targets of interest. For clinical cases involving a lesion in deep white matter structures, the population probability quantified by the tract-to-region connectome can provide the likelihood of an affected white matter pathway leading to functional deficits of a cortical region. Conversely, in fMRI, EEG, or SEEG studies identifying a cortical region of interest, the tract-to-region connectome can translate the findings to their corresponding white matter pathways based on population probabilities. Subsequently, the information can delineate the circuit mechanism behind a cognitive model or verify the structure-function hypotheses. Both of them may

further help explore new white matter targets for neurological modulations using deep brain stimulation, focused ultrasound ablation, or laser ablation.

Comment 2:

The methodology of the work seems sound, but is in parts described very superficially. Particularly the automated tractography is not reproducible with the information provided in the manuscript.

Responses:

This is indeed a shortcoming of the manuscript. The Method section is substantially revised to include paragraphs and provide the details.

Modifications Made:

[Methods Section]

Automated tractography

For each subject, 52 white matter bundles were mapped using the automated tractography pipeline in DSI Studio (<http://dsi-studio.labsolver.org>), which combined deterministic fiber tracking algorithm⁶³, parameter saturation and randomized parameters, 12 topology-informed pruning⁶⁴, and trajectory-based tract recognition¹² (detailed in the next section) as an integrated interface. The default settings were used: the anisotropy threshold was uniformly and randomly selected from 0.5 to 0.7 Otsu threshold. The angular threshold was uniformly and randomly selected from 15 to 90 degrees. The step size was uniformly and randomly selected from 0.5 to 1.5 voxel spacing. The minimum length was 30 mm. The tracking was repeated until the ratio of streamlines to voxels reached 1.0.

Trajectory-based tract recognition

The trajectory recognition was based on a population-averaged tractography atlas 28 in the 2009 ICBM152 nonlinear asymmetry space (publicly available at <https://brain.labsolver.org>), and the nearest neighbor method was used to classify the subject tracts. Specifically, the atlas trajectories were nonlinearly warped to subject space based on the HCP-1065 anisotropy images, which imposed higher weights in core white matter structures due to their high anisotropy value. The Hausdorff distance⁶⁵ was computed between each subject tract and each atlas tract. The shortest distance then determined the label of subject tracts. Some tracts might substantially deviate from all atlas trajectories, and thus a maximum allowed Hausdorff distance (termed tolerance distance) of 16 was used to remove them. After recognition, each bundle was pruned by topology-informed pruning⁶⁴. If the pruning yielded no result, the minimum Hausdorff distance was increased to 18 and 20 mm, and the recognition-pruning step was repeated. After increasing the minimum Hausdorff distance, there was a possibility that the automated tractography yielded no result. In this study, 4 out of the 52 bundles did not reach 100% yield rate for all 1065 subjects: left corticobulbar tracts (983/1065), right corticobulbar tract (1054/1065), right arcuate fasciculus (1046/1065), right occipital corticopontine tract (1064/1065). The missing of the corticobulbar and corticopontine tracts in some subjects could be due to the limitation of the fiber tracking algorithm to capture substantial turning of the pathways. The right arcuate fasciculus in some subjects was entirely labeled as SLF III due to no connection to the superior temporal lobe.

The steps mentioned above, including subject-space fiber tracking, parameter saturation, randomized parameters, topology-informed pruning, anisotropy-based warping, Hausdorff distance computation, were integrated into DSI Studio and available in both the command line and graphical user interface. The source code is also available on GitHub repository at <https://github.com/frankyeh/DSI-Studio>.

Comment 3:

While the conducted experiments are sound, several questions arose while studying the manuscript that are not covered by the experiments and also not discussed:

- The author mentions that around 15% of the observed tract to region connections occur often but not always (likelihood between 5% and 95%) and attribute this to "substantial individual variation". This is the case for all used cortical parcellations. This tells me that the cause might not be "individual variation" but maybe simply caused by some factor in the tractography pipeline, such as noise and an insufficient resolution, that causes tracts that actually only connect to N regions to randomly "leak" to neighboring regions. Whatever the reason may be, it would be interesting to look into this further. One way to gain more information about this, apart from actually looking at these cases, could be the use of random but parameterizable parcellations instead of actual atlas-based parcellations.

Responses: This is an interesting approach to examine the robustness of results. Additional analysis was conducted using a random parcellation revised from Craddock (2011) to the ICBM152 gray matter, as shown below (added as Supplementary Fig 1a).

The generated tract-to-region connectome shows 87.09% of entries (8152/9360) with a likelihood between 5% and 95%. (added as Supplementary Fig 1b). This is remarkably similar to 86.5% reported in HCP-MMP, 85.45% in Brodmann, and 84.93% in Kleist.

The Methods and Results section are revised with one new figure added as Supplementary Figure 1.

Modifications Made:

[Introduction Section]

Four tract-to-region connectome matrices were quantified using the Brodmann parcellation, Kleist parcellation¹³, the Human Connectome Project's multimodal parcellations (HCP-MMP)¹⁴, and a random parcellation, respectively.

[Supplementary Fig.1 added]

Shown above

[Results Section]

A random parcellation with 360 regions derived from Craddock's random parcellations¹⁸ also showed a similar result (Suppl. Fig. 1): 87.09% of the matrix entries (8152 out of 9360) had probability values ranging between 95% and 5%. Overall, the tract-to-region connectome showed that the young adult population shares a similar connective pattern in ~85% of the tract-to-region entries. The remaining ~15% entries have substantial individual variations with population probability between 5% and 95%, thus warranting individualized mapping.

Comment 4:

- In the approach presented by the authors, the information which brain region is connected to which other region is lost. Instead of completely skipping this information, it would have been straightforward to extend classic region to region connectivity with the tract specific information included in the presented tract to region connectivity, e.g. representable by a 3D connectivity matrix ("region to region via tract connectivity"). Why did the author choose to discard this information?

Responses: Thank you for bringing up this significant contribution.

We have previously reported tract-wide region-to-region connectivity matrices and constructed 2D+ "t" connectograms for each tract at different categorical levels (shared at brain.labsolver.org) in the 2018 study (Yeh et al. 2018):

(Yeh, Nueorimage, 2018)

The data shared in this study also include the tract coverage that can also derive the region-to-region connectome. The manuscript is revised to highlight this impact.

Modifications Made:

[Discussion Section]

The tract-to-region connectome matrices and all intermediate data, such as tractogram of each bundle, read-to-track subject data, and source code, are publicly available at <https://brain.labsolver.org>. The shared data could also construct the conventional region-to-region connectome for each white matter pathway, as illustrated in our previous connectome study²⁸.

Comment 5:

- The author only mentions the tract atlas used for the automated tractography does not describe it in detail. Could the choice of this atlas affect the results? Are there alternatives? Why was this atlas chosen? This should be discussed.

Responses:

Yes, the choice of the atlas would affect the result.

However, the challenge is that existing white matter atlases were either (1) NIFTI files of voxel-based masks or (2) trajectories in individual subjects. This study would need an atlas that (1) provides TRK files recording the trajectories of white matter pathways, and (2) is aggregated or averaged from a population to minimize the concern of individual differences or bias.

To our best knowledge, the only atlas that meets these requirements is the population-average tractography atlas constructed by the author (Neuroimage . 2018 Sep;178:57-68), currently used in this study.

The following figure shows the differences.

Left: *trajectory-based* atlas used in this study

Right: *voxel-based* atlas from Catani (2008)

The Discussion section is revised, with one conclusion paragraph added to discuss this critical point.

Modifications Made:

[Discussion Section]

Because of several practical considerations, we could not comprehensively explore how a different tractography atlas or a segmentation tool affects the tract-to-region connectome. Most tract segmentation tools used *region-based* recognition. They relied on cortical parcellations^{8,55} or end regions of tracts¹⁰ to recognize a tract. This approach would interfere with the tract-to-region interpretation and lead to circular findings in the tract-to-region connectome. This study customized a *trajectory-based* recognition method to recognize each trajectory without resorting to tract clustering, which could also override the clustering analysis applied to the connectome. In addition to limitations in tools, existing atlases^{56,57} were either *voxel-based* or annotated in individuals, as summarized in a recent study⁹. The *voxel-based* atlases provided only masks of tract coverage and did not have trajectory data needed by trajectory-based recognition. Constructing the tract-to-region connectome required a *tract-based* atlas aggregated from a population in the ICBM152-space to minimize individual differences or bias²⁸, and not all atlases are fit for this purpose. The one used in this study was scrutinized to satisfy these requirements.

Nonetheless, studies have shown that the tractography segmentation could be different due to anatomical views⁵⁸ or segmentation tools⁵⁹. The differences due to tools were direct results of different input data: *region-based* recognitions used cortical parcellations, while *trajectory-based* recognition used the topology of white matter tracts. On the other hand, the differences in anatomical views were mainly due to discrepancies in the existing categorical systems and tract nomenclature^{58,60}. While there is no ambiguity in the anatomical locations of white matter pathways, much of the recent disputes focus on detailed subcomponents and their categorical relations⁶¹. Resolving them would need new neuroanatomical evidence from tract-tracing or cadaver dissection studies. The population-based tractography and its corresponding tract-to-region connectome are thus subject to future revisions and updates to ensure their up-to-date accuracy.

Comment 7:

- The author describes that a tract and a region are regarded as connected as soon as their binary masks overlap. In this scenario it seems possible that regions connect to parts of the tracts that are actually not endpoints. This should be discussed.

Overall the work is well presented and the approach seems very interesting for the research community and I recommend acceptance of the manuscript if the above issues and comments are addressed.

Responses:

This is indeed a critical limitation in tractography. The manuscript is revised to emphasize that fiber tracking could not confirm the existence of a synaptic connection.

Modifications Made:

[Discussion section]

Moreover, for bundles mapped in this study, the tract-to-region relation was determined by a simple overlap in the binary mask, and there could be false-positive results. On top of these limitations, the included tracts were also subject to errors such as trajectory deviation, premature termination, incorrect routing, as discussed in a recent review⁵⁴. Although multiple strategies have been leveraged to reduce false connections, we cannot rule out possible errors in the current form of the tract-to-region connectome.

Reviewer #3:

General Comment:

In this paper the author present a new way to present and analyse structural connectome data of the human brain. The paper is well written and figures, in particular, are very clear and easy to understand. However, I'm not sure I fully understand what is the main outcome of this study.

Responses:

Thank you for the objective criticisms, which are very helpful in improving the study and correcting the shortcomings. The unsupported conjectures about tract clustering were removed from the Results section and discussed in the Discussion section. The manuscript has been substantially revised to correct questionable arguments, provide more methodological details, add supporting references, disclose possible pitfalls, and emphasize the room for future improvements.

The following is point-to-point responses:

Comment 1:

Segmenting white matter tract using automatic and hierarchical clusters methods has been presented in several previous works. The novelty here is that a connectome-like analysis is applied to a predefined set of tracts extracted using an automatic approach. I fully agree with the author that connectomes built using only a region-to-region approach may not provide enough information of the real underlying white matter connectivity and may lead to a "black-box" analysis approaches. However, the usual definition of connectome also implies that we are also trying to look at the entirety of white matter connections (within data resolution limits, of course). However, by selecting only a subset of white matter tracts as provided by the automatic segmenting tool I don't think we are getting an unbiased connectome view and everything will be highly depended on how these tracts are extracted. Moreover, all classical network properties will be biased by the actual number and selections of tracts. Change the set of tracts or how they are extracted and the topology will likely be different.

Responses:

This is indeed a limitation of the tract-to-region connectome in this study. The connectome mapped here is not a complete connectome that includes all pathways. The revised manuscript also discusses the specific reasons for excluding certain pathways. Moreover, the tract-to-region connectome is a bipartite graph, and even if a complete mapping were achieved, not all classical network properties would be applicable. This precaution is also added to the revision.

The Discussion section is revised to include this limitation and give precautions of the applicable scopes.

Modifications Made:

[Discussion section]

There are limitations in this study. The tract-to-region connectome did not include cerebellar, commissural, brainstem, or connections between subcortical structures. Excluding cerebellar and brainstem pathways are due to different slices coverage near the brainstem and insufficient spatial resolution to generate reliable fiber tracking results. The commissural pathways are excluded because of the limited ability of the fiber tracking methods to resolve crossing-kissing patterns when the corpus callosum crosses the hemispheres⁵³.

[Discussion section]

Further, the existing graph-theoretical analysis⁵ may not be readily applicable to the tract-to-region connectome because the tract-to-region concept is conceptually different from the conventional region-to-region one. While the region-to-region connectome implies an undirected graph, the tract-to-region matrix appears to be a subset of an undirected graph called a *bipartite graph*. The bipartite graph comprises two disjoint sets of nodes, one for tracts and one for regions. The nodes representing tracts could not be equally exchanged with nodes

representing regions. Since most network measures view all nodes equally in the computation, applying these network analyses to the tract-to-region connectome could lead to questionable results. Further theoretical development is thus required to translate network measures to the tract-to-region connectome.

Comment 2a:

Not much information is given about how these tracts are selected and extracted and I think this the main factor driving all the results in the manuscript. The author should significantly expand this part. Per se, the tract-to-region approach is more a way to very nicely display how presegmented tracts connect different brain regions.

Responses:

Thank you for pointing out this major shortcoming of the manuscript. The Methods section is substantially revised, with details added to describe how the tracts were generated.

Modifications Made:

[Methods Section]

For each subject, 52 white matter bundles were mapped using the automated tractography pipeline in DSI Studio (<http://dsi-studio.labsolver.org>), which combined deterministic fiber tracking algorithm⁶⁵, randomized parameter saturation¹², topology-informed pruning⁶⁶, and trajectory-based tract recognition¹² (detailed in the next section) as an integrated interface. The default settings were used: the anisotropy threshold was uniformly and randomly selected from 0.5 to 0.7 Otsu threshold. The angular threshold was uniformly and randomly selected from 15 to 90 degrees. The step size was uniformly and randomly selected from 0.5 to 1.5 voxel spacing. The minimum length was 30 mm. The tracking was repeated until the ratio of streamlines to voxels reached 1.0.

The tract recognition uses the nearest neighbor method to classify the subject tracts. The recognition used a population-averaged tractography atlas²⁸ aggregated from the young adult population and vetted by a team of neuroanatomists without resorting to cortical parcellation. A new version of the atlas in the ICBM152 nonlinear asymmetry space (publicly available at <https://brain.labsolver.org>) was used in this study. For each subject, the tractography atlas was nonlinearly warped to subject space using the diffeomorphic mapping derived between the subject's anisotropy image and the ICBM152-space anisotropy image. The Hausdorff distance⁶⁷ was computed between each subject and atlas tract. The shortest distance then determined the label of subject tracts. Some tracts might substantially deviate from all atlas trajectories, and thus a maximum allowed Hausdorff distance (termed tolerance distance) of 16 was used to remove them. The minimum Hausdorff distance was increased to 18 and 20 mm if no bundle was found after topology-informed pruning⁶⁶. In this study, 4 out of the 52 bundles did not reach 100% yield rate for all 1065 subjects: left corticobulbar tracts (983/1065), right corticobulbar tract (1054/1065), right arcuate fasciculus (1046/1065), right occipital corticopontine tract (1064/1065). The missing of the corticobulbar and corticopontine tracts in some subjects could be due to the limitation of the fiber tracking algorithm to capture substantial turning of the pathways. The right arcuate fasciculus in some subjects was entirely labeled as SLF III due to no connection to the superior temporal lobe.

Comment 2b: (continue within the same paragraph)

A big concern I have here is that depending on the segmenting algorithm most of the tracts may be already defined, in first place, by cortical regions making the analysis presented in this manuscript a circular reasoning problem. The constant sharp 95%/5% of connectivity ratio for some of the tract-to-region makes me suspect this can be the case.

Responses:

Totally agree that *circular analysis* is a critical point.

This study did consider *circular analysis* problem in the first place and thus customized a trajectory-based recognition tool that avoided approaches with circular analysis concerns, such as using cortical parcellation, end regions of tracts, or applying clustering before recognition.

The Introduction and Discussion sections are substantially revised to explain the circular analysis issue.

Modifications Made:

[Introduction Section]

Several technical advances were used to construct the tract-to-region connectome (Fig. 1). The white matter bundles of the 1065 young adults were mapped using automated tractography. *Although many automated tractography methods are available⁶⁻¹¹, most have used cortical parcellation to recognize tracts. These region-based methods could lead to circular analysis in the tract-to-region connectome. Therefore, this study used trajectory-based recognition¹² and did not filter tractogram by brain regions.* After trajectory-based recognition, connections substantially deviated from the expert-vetted tractography atlas were removed.

[Discussion section]

Because of several practical considerations, we could not comprehensively explore how a different tractography atlas or a segmentation tool affects the tract-to-region connectome. Most tract segmentation tools used region-based recognition. They relied on cortical parcellations^{8,55} or end regions of tracts¹⁰ to recognize a tract. This approach would interfere with the tract-to-region interpretation and lead to circular findings in the tract-to-region connectome. This study customized a trajectory-based recognition method to recognize each trajectory without resorting to tract clustering, which could also override the clustering analysis applied to the connectome.

Comment 3:

The use of different cortical parcellations only in part can solve this problem because of the significant overlap of many of the major cortical regions. A more appropriate analysis would have been to compare different segmentation tools and show if and how the connectome topology modify itself. If the tract-to-region approach is providing meaningful results, a consistent pattern should emerge.

Responses:

In a previous collaboration study by Schilling et al. (<https://doi.org/10.1016/j.neuroimage.2021.118451>), we worked with Schilling's team to compare different segmentation tools and confirmed your insight: *there were substantial differences across tools.*

The differences are due to whether they used cortical segmentation to assist tract segmentation (thus would have circular analyses concern if used in this study) or whether they applied tracts clustering before segmentation (would also have circular results in clustering analysis). The tool used in this study was specially customized to avoid using cortical segmentation and before-segmentation clustering, whereas other tools (e.g., XTRACT, TrackSeg) did not avoid using cortical regions or clustering and thus might not be ideal to construct the tract-to-region connectome due to circular analysis concerns.

The Discussion sections were revised to emphasize this point, with cautions on the fact that different tools would lead to different results:

Modifications Made:

[Discussion section]

Because of several practical considerations, we could not comprehensively explore how a different tractography atlas or a segmentation tool affects the tract-to-region connectome. Most tract segmentation tools used region-

based recognition. They relied on cortical parcellations^{8,55} or end regions of tracts¹⁰ to recognize a tract. This approach would interfere with the tract-to-region interpretation and lead to circular findings in the tract-to-region connectome. This study customized a *trajectory-based* recognition method to recognize each trajectory without resorting to tract clustering, which could also override the clustering analysis applied to the connectome.

[Discussion section]

Nonetheless, studies have shown that the tractography segmentation could be different due to anatomical views⁵⁸ or segmentation tools⁵⁹. The differences due to tools were direct results of different input data: *region-based* recognitions used cortical parcellations, while *trajectory-based* recognition used the topology of white matter tracts.

Comment 4:

Looking at the clustering results presented here I'm surprised to read that the SLFI is part of the Cingulum system. The Cingulum is very well anatomically defined and it is running within the cingulate gyrus or surrounded by the cingulate cortex. On the contrary SLF1 is more lateral-superior and connects superior frontal lobe with mostly the superior parietal lobule and other parietal regions. Again, depending on how the tracts are defined/extracted here, some of the termination may be shared but not all tracts have only two clear termination points. Some have multiple branches and some like the cingulum may multiple U-shape fibres running within the tract itself. Similarly, while SLF3 can be associated to the arcuate system I'm not sure SLF2 can be said to belong to anatomically or functionally to the arcuate system. I can't find justification of that in the references provided. Also I don't think that merging together IFOF with Uncinate as a system, just because they share the same frontal region, is a way to improve our understanding on human brain anatomy.

Responses:

The manuscript is revised to remove unsupported claims from the Results section. We also revised the manuscript to emphasize that the clustering results only offer a different perspective on white matter tract relations. By no means could the results be used to change naming conventions or modify the definition of neuroanatomy. A detailed discussion and supporting references were added to the Discussion section.

Modifications Made:

[Result section]

Fig. 9 further shows the similarity matrix between the association pathways (Fig. 9a), and the dendrogram illustrates the hierarchical clustering results (Fig. 9b) based on the HCP-MMP tract-to-region connectome. The left and right hemispheres show highly similar hierarchical relations that groups association pathways into four systems, including the arcuate system (purple), anterior ventral system (red), posterior ventral system (cyan), and cingulum system (green). The first system includes AF, SLF II, SLF III, and FAT. These pathways all connect to Broca's area and have correlated with language functions shown by several studies (detailed in the Discussion section). The second system includes MdLF, TPAT, VOF, and ILF. TPAT has several alternative naming, such as the posterior AF, posterior SLF, or SLF-TP (Suppl. Table 1). The third system includes UF and IFOF, and both are characterized by their frontal connection from the temporal and occipital lobes, respectively. The fourth system includes all cingulum pathways and SLF I, likely because the SLF I is closely adjacent to the cingulum at (Y=-3 and Y=-11) and entirely separated from SLF II and III by FAT (Fig. 3b). The above data-driven clustering results showed the relation between white matter pathways based on their similarity in the tract-to-region connectome.

[Discussion section]

We also derived the hierarchical relation between white matter bundles. While many studies have been conducted to cluster white matter tracts⁴⁰⁻⁴⁵, these clustering methods did not consider the connective pattern with the cortical regions. The clustering in this study showed that SLF II and SLF III are closely related to AF, whereas SLF I is closely related to the cingulum. These results may appear questionable and astonishing at first

glance, but there are supporting references: Catani et al.²⁰ showed SLF II and III as the *anterior segment* of the AF, which did not include SLF I. Wang et al.⁴⁶ suggested that the SLF I could be viewed as part of the cingulum system. From the clinical perspective, especially in the surgical intervention of brain tumors, the neurosurgical consensus is that the *eloquent area* correlated with post-surgical functional deficits includes regions innervated by AF, SLF II, and SLF III⁴⁷⁻⁴⁹. These areas did not include SLF I because SLF I did not show significant language function⁵⁰. Furthermore, SLF I was first delineated by anterograde tract-tracer technique in rhesus macaques⁵¹, where detailed mapping was illustrated by Schmahmann and Pandya's work⁵². In their work, among 15 cases enhancing the SLF I (case 1, 2, 3, 4, 6, 7, 9, 17, 22, 26, 27, 28, 29, 31, 33), 12 of them (except case 26, 27, 28) also enhanced the cingulum bundle as labeled by the authors. In comparison, only two cases (case 4 and 31) enhanced SLF II, and three (case 6, 7, 33) enhanced SLF III. This hinted a closer relationship between the SLF I and cingulum than with the SLF II or SLF III. Nonetheless, it is noteworthy that the clustering in this study was based on the tract-to-region connectome entries, and by no means could this be used to confirm a new naming convention or provide a new neuroanatomical definition as each bundle has well-defined anatomical locations, as shown in Fig. 2. More functional or lesion-based studies are needed to support or refute these clustering results.

Comment 5:

I think that while some of these anatomical claims are interesting, some are also misleading and driven by how the specific set of tracts have been dissected. I think that strong anatomical claims should not just coming from a pure data-driven approach but they need to strongly supported and validated using independent anatomical or clinical data.

Responses:

Totally agree with this insight.

The manuscript is revised to include the limitation of the data-driven approach, and a conclusion paragraph is added to emphasize the importance of other tissue validation approaches.

Modifications Made:

[Discussion section]

Moreover, for bundles mapped in this study, the tract-to-region relation was determined by a simple overlap in the binary mask, and there could be false-positive results. On top of these limitations, the included tracts were also subject to errors such as trajectory deviation, premature termination, incorrect routing, as discussed in a recent review⁵⁴. Although multiple strategies have been leveraged to reduce false connections, we cannot rule out possible errors in the current form of the tract-to-region connectome.

[Discussion section]

On the other hand, the differences in anatomical views were mainly due to discrepancies in the existing categorical systems and tract nomenclature^{58,60}. While there is no ambiguity in the anatomical locations of white matter pathways, much of the recent disputes focus on detailed subcomponents and their categorical relations⁶¹. Resolving them would need new neuroanatomical evidence from tract-tracing or cadaver dissection studies. The population-based tractography and its corresponding tract-to-region connectome are thus subject to future revisions and updates to ensure their up-to-date accuracy.

REVIEWER COMMENTS

Reviewer #1 (Remarks to the Author):

The manuscript is significantly clarified and improved. I do have some minor comments regarding the updates related to justification of the choice of tract identification method and atlas employed.

The statement that there are exist no population aggregated tractography atlases is misleading. The data used to train TractSeg are publicly available (curated tracts in a population of subjects), and a 100-subject tractography atlas (with clusters labeled with the tracts to which they belong) is publicly available from Zhang, Fan, et al. "An anatomically curated fiber clustering white matter atlas for consistent white matter tract parcellation across the lifespan." *NeuroImage* 179 (2018): 429-447. Additional tractography atlases are likely available as well. Furthermore, many automated methods for identifying tracts from tractography are available, have been trained on tractography atlases, and would serve the same purpose as the methods employed in this manuscript (identifying sets of tracts without using cortical parcellation data). Example methods that can be directly employed for this purpose include RecoBundles in dipy, whitematteranalysis, TractSeg, other recently proposed deep learning methods, etc. I recommend a more nuanced defense of the particular choice of atlas and tract identification method used in this paper, rather than trying to claim it's the only identification method that is trajectory based or that it's the only atlas with trajectories in it.

I am confused about the statements that tract clustering before segmentation would have circular results. In my recollection, the atlas used in this paper is the one in DSIstudio, which used clustering to initially group fibers for viewing and labeling by neuroanatomists. In this way, similar to other atlases, clustering was used to help present a large amount of streamline data to experts. Does this mean the analysis is circular? Personally I don't think so, but the authors' phrase "resorting to tract clustering" makes it sound like the authors think clustering is a bad idea, when they have used it to create their atlas and also applied it in this paper to summarize their connectome results. I am left confused by the seemingly contradictory statements on this topic of clustering.

These comments are minor and hopefully can be easily clarified and improved. Of course, including results from another tract segmentation method, similar to the nice way tractography bottlenecks were recently studied using two sets of identified tracts (citation below), would solve all these issues in some sense. I'm not sure this extra work is needed, as the main point of the current manuscript is a new perspective on how to study the connectome. Hopefully the anatomical claims have now been softened enough that the dependence of results on a particular tract segmentation methodology is not of primary importance to the overall message. However, it should be clear from the Discussion/Conclusion that the primary result is this new way of thinking about the connectome rather than the initial findings that are

specific to one tract identification pipeline. Here's the recent paper that used two tract identification methods, as a reference:

Schilling, Kurt G., et al. "Prevalence of white matter pathways coming into a single white matter voxel orientation: The bottleneck issue in tractography." *Human brain mapping* 43.4 (2022): 1196-1213.

Reviewer #2 (Remarks to the Author):

The authors addressed most comments and questions raised in my first review to my full satisfaction. I have two more followup-comments on my previous comments 1 and 5:

(1) The authors extended their description of possible applications where their approach could yield valuable new information. Intuitively this is sound, but it would be desirable to actually apply the method in a larger study to actually show the impact. This might very well be beyond the scope of this work and I do think the presented method is worth publishing without this.

(5) The authors mention that no atlas is available that satisfies the conditions to include bundle masks as well as streamline-based tracts. What about tract segmentation approaches such as TractSeg (<https://github.com/MIC-DKFZ/TractSeg/>)? This method is state of the art, yields tract masks, streamline bundles and the underlying neural network is trained on a larger collective of subjects. If this method does indeed satisfy all requirements, it would be interesting to analyze what effect the different method has on the overall results.

Reviewer #3 (Remarks to the Author):

Thanks for the comments and the changes of the new version. Overall I think the work has improved but there are still some points that need to be clarified or corrected.

Following the author answers, it is now stated that this method is not about a complete "connectome" analysis but it is looking at a subset of tracts. Also because of the topology of the network is different (i.e. a bipartite graph vs uni-directed graph) conventional network analysis techniques are precluded. To avoid confusion in terminology I would suggest to refer to this method as a way to look more at "brain networks" than the connectome.

Following comments to reviewer 2 I realise that the way tracts are associated to a region can be a bit problematic. In my view only terminations of the tracts should be considered. I understand your point about the inability to separate mono-synaptic from multi-synaptic connections but this is not solving the practical issue of simply overlapping a tract with a cortical region. A long superficial associative tract will be connected with all nearby cortical regions even if the main termination points are much smaller and distant. E.g. The IFOF will be connected with the insula just because it is passing nearby. Is this intentional or a limitation of the study?

Catani et al. did not associate SLF2 to the anterior segment of the arcuate fasciculus but only SLF3. Please correct or remove the sentence.

Reviewer #1

Comment:

The manuscript is significantly clarified and improved. I do have some minor comments regarding the updates related to justification of the choice of tract identification method and atlas employed.

Responses:

Thank you for pointing out misleading statements in the manuscript. The manuscript is revised accordingly.

Comment1a:

*The statement that there are exist no population aggregated tractography atlases is misleading. The data used to train TractSeg are publicly available (curated tracts in a population of subjects), and a 100-subject tractography atlas (with clusters labeled with the tracts to which they belong) is publicly available from Zhang, Fan, et al. "An anatomically curated fiber clustering white matter atlas for consistent white matter tract parcellation across the lifespan." *NeuroImage* 179 (2018): 429-447. Additional tractography atlases are likely available as well.*

Responses:

Thank you for pointing out the misleading statement, and the manuscript is revised to clarify the details.

To summarize:

The limitation is on the nearest-neighbor recognition used in this study, which cannot effectively utilize voxel-based atlases. Another limitation of the nearest neighbor classifier used in this study is that it is susceptible to individual variations and tends to overfit. The individual streamline would need a group average to avoid the shortcomings. In Zhang's work, several outstanding atlases are summarized in Table 1. Most of them are voxel-based atlases and thus cannot be used in this study. The rest atlases were individual streamlines, and they would need adequate averaging to be effectively utilized by the nearest neighbor recognition method.

Again, this is a limitation of the nearest neighbor classifier due to its simple implementation. The applicable atlas is intrinsically limited. An ultimate solution is to develop a more robust, universal recognition approach that is less sensitive to individual differences and applicable to voxel-based and streamline-based atlases.

The Method section is revised to add this rationale. The Discussion section is also updated to avoid misleading statements.

Modifications Made:

Method section

The tract recognition used the nearest neighbor method to classify tracts. Since the nearest neighbor classifier is sensitive to noisy data and prone to overfitting, training data preparation was critical for best performance. Most population-based atlases have substantial individual variations and thus would require additional averaging to minimize individual differences. Therefore, in this study, the recognition used a population-averaged tractography atlas²⁸, which was aggregated from the young adult population and vetted by a team of neuroanatomists. An updated version of the atlas in the ICBM152 nonlinear asymmetry space (publicly available at <https://brain.labsolver.org>) was used in this study.

Discussion section

A solution is to use white matter trajectories to recognize tracts, but trajectories-based recognition methods may not effectively utilize all existing atlases^{9,56,57}. Most white matter atlases were voxel-based volumes that provided only masks of tract coverage and did not have trajectory coordinates needed by trajectory-based recognition. Few atlases provide trajectories coordinates for individual subjects, but a group average would be needed to minimize the individual differences. This averaging step is critical for classifiers that are sensitive to noisy data.

Comment1b:

Furthermore, many automated methods for identifying tracts from tractography are available, have been trained on tractography atlases, and would serve the same purpose as the methods employed in this manuscript (identifying sets of tracts without using cortical parcellation data). Example methods that can be directly employed for this purpose include RecoBundles in dipy, whitematteranalysis, TractSeg, other recently proposed deep learning methods, etc. I recommend a more nuanced defense of the particular choice of atlas and tract

identification method used in this paper, rather than trying to claim it's the only identification method that is trajectory based or that it's the only atlas with trajectories in it.

Responses:

Thank you for pointing out this key point. The manuscript is revised to avoid giving a false impression that there is only one tract identification method.

Modifications Made:

Discussion section

The tract-to-region relation could be derived using different tools or atlases, but additional customizations may be needed to address unique technical concerns when deriving the tract-to-region relation. Specifically, tract segmentation tools often used cortical parcellations^{8,55} or end regions of tracts¹⁰ to recognize a tract. The tracts defined by cortical parcellation would show connections according to the supplied cortical parcellations, leading to circular results in the tract-to-region connectome.

Comment 2a:

I am confused about the statements that tract clustering before segmentation would have circular results. In my recollection, the atlas used in this paper is the one in DSIstudio, which used clustering to initially group fibers for viewing and labeling by neuroanatomists. In this way, similar to other atlases, clustering was used to help present a large amount of streamline data to experts. Does this mean the analysis is circular? Personally I don't think so, but the authors' phrase "resorting to tract clustering" makes it sound like the authors think clustering is a bad idea, when they have used it to create their atlas and also applied it in this paper to summarize their connectome results. I am left confused by the seemingly contradictory statements on this topic of clustering.

Responses:

I agree with your point, and the sentence is removed to avoid this questionable statement.

Modifications Made:

(Statement removed)

~~This study customized a trajectory-based recognition method to recognize each trajectory without resorting to tract clustering, which could also override the clustering analysis applied to the connectome.~~

Comment 2b:

These comments are minor and hopefully can be easily clarified and improved. Of course, including results from another tract segmentation method, similar to the nice way tractography bottlenecks were recently studied using two sets of identified tracts (citation below), would solve all these issues in some sense. I'm not sure this extra work is needed, as the main point of the current manuscript is a new perspective on how to study the connectome. Hopefully the anatomical claims have now been softened enough that the dependence of results on a particular tract segmentation methodology is not of primary importance to the overall message. However, it should be clear from the Discussion/Conclusion that the primary result is this new way of thinking about the connectome rather than the initial findings that are specific to one tract identification pipeline. Here's the recent paper that used two tract identification methods, as a reference:

Schilling, Kurt G., et al. "Prevalence of white matter pathways coming into a single white matter voxel orientation: The bottleneck issue in tractography." Human brain mapping 43.4 (2022): 1196-1213.

Responses:

Thank you for the suggestion, we add a conclusion to emphasize the main focus of this study, which is the new perspective to study the connectome.

The anatomical arguments are softened as suggested.

Modifications Made:

Discussion section:

This study mainly focuses on the concept of the tract-to-region connectome, and the variations due to different tracking recognition tools were not investigated. The tract-to-region relation could be derived using

different tools or atlases, but additional customizations may be needed to address unique technical concerns when deriving the tract-to-region relation.

Discussion section:

Despite those discrepancies, it is noteworthy that the anatomical locations of white matter pathways are well-defined, and this study did not invent new white matter structures. The pathways mapped in this study (e.g., those shown in Fig. 2 and Fig. 3) are anatomically consistent with existing atlases from other tools and studies, and the clustering results reported were consistent across three different cortical parcellations.

Reviewer #2

The authors addressed most comments and questions raised in my first review to my full satisfaction. I have two more followup-comments on my previous comments 1 and 5:

Comment 1:

(1) The authors extended their description of possible applications where their approach could yield valuable new information. Intuitively this is sound, but it would be desirable to actually apply the method in a larger study to actually show the impact. This might very well be beyond the scope of this work and I do think the presented method is worth publishing without this.

Responses:

Thank you for the comment. Future work on larger data is undergoing, and we have shared preprocessed, ready-to-track data at <https://brain.labsolver.org>, which includes HCP-aging, HCP-development, and developing HCP. The instructions for automatic tractography are detailed at https://dsi-studio.labsolver.org/doc/gui_t3_atk.html, with tutorial videos available. DSI Studio can track the data to construct the tract-to-region connectome in aging and developmental populations.

We added these potential future works in the Discussion section to invite participation from the community.

Modifications Made:

Discussion section:

This consistency may support future works extending tract-to-region mapping to lifespan studies. To this end, ready-to-track data for HCP-aging, HCP-developmental, and developing HCP studies and sample processing scripts are available at <https://brain.labsolver.org> to assist further brain mapping endeavors.

Comment 2:

(5) The authors mention that no atlas is available that satisfies the conditions to include bundle masks as well as streamline-based tracts. What about tract segmentation approaches such as TractSeg (<https://github.com/MIC-DKFZ/TractSeg/>)? This method is state of the art, yields tract masks, streamline bundles and the underlying neural network is trained on a larger collective of subjects. If this method does indeed satisfy all requirements, it would be interesting to analyze what effect the different method has on the overall results.

Responses:

TractSeg uses "end region segmentation" (shown in figures at <https://github.com/MIC-DKFZ/TractSeg/>) that may have a circular analysis concern. This concern could be lifted if the pipeline was modified to remove this component. Although we did not test other tools, the tract mapping results (e.g. Fig. 2 and Fig. 3) are highly consistent with those from TractSeg and others. The consistency could imply reproducible results using other mapping approaches.

The Discussion section is revised to explain this.

Modifications Made:

Discussion section:

The pathways mapped in this study (e.g., those shown in Fig. 2 and Fig. 3) are anatomically consistent with existing atlases from other tools and studies, and the clustering results reported were consistent across three different cortical parcellations. The tract-to-region relation could be derived using different tools or atlases, but additional customizations may be needed to address unique technical concerns when deriving the tract-to-region relation. Specifically, tract segmentation tools often used cortical parcellations^{8,55} or end regions of tracts¹⁰ to recognize a tract. The tracts defined by cortical parcellation would show connections according to the supplied cortical parcellations, leading to circular results in the tract-to-region connectome.

Reviewer #3:

Thanks for the comments and the changes of the new version. Overall I think the work has improved but there are still some points that need to be clarified or corrected.

Comment 1:

Following the author answers, it is now stated that this method is not about a complete "connectome" analysis but it is looking at a subset of tracts. Also because of the topology of the network is different (i.e. a bipartite graph vs uni-directed graph) conventional network analysis techniques are precluded. To avoid confusion in terminology I would suggest to refer to this method as a way to look more at "brain networks" than the connectome.

Responses:

The manuscript is screened and revised to remove "connectome" if applicable.

I would request to keep the term "tract-to-region connectome" even though it is not a complete one.

The following are some supporting views:

1. Although this study did not map the entire connectome, the tract-to-region connectome still conceptually exists beyond methodological limitations and could be realized by different brain mapping methods.
2. The first connectome studies by Patric Hagmann et al. (2007) or Olaf Sporns et al. (2007) also did not include the cerebellum and brainstem. It seems that partial mapping of the connectome, depending on the completeness of the mapping, could still be called the connectome.
3. Replacing the "tract-to-region connectome" with "tract-to-region brain networks" could be confusing.

Modifications Made:

(track-editing attached with the clean manuscript)

2 modifications in the Abstract

2 modifications in the Introduction section

Comment 2:

Following comments to reviewer 2 I realise that the way tract are associated to a region can be a bit problematic. In my view only terminations of the tracts should be considered. I understand your point about the inability of separate mono-synaptic from multi-synaptic connection but this is not solving the practical issue of simply overlapping a tract with a cortical region. A long superficial associative tract will be connected with all nearby cortical regions even if the main termination points are much small and distant E.g. The IFOF will be connected with the insula just because it is passing nearby. Is this intentional or a limitation of the study?

Responses:

It is both intentional and a limitation of the study.

The intention is to address the "gyral bias" of deterministic tractography, which showed only connections to the tip of the gyrus and ignored those to the "gyral bank." My experience using termination often resulted in too conservative results, and many well-known connections would not be captured. The limitation, as you mentioned, is spurious connections such as the IFOF connection to Pol1 and 52, which is due to IFOF passing within the resolution limit. Unfortunately, tractography cannot assert synaptic connections. This limitation is added to the Discussion.

Modifications Made:

This setting was used to compensate for tractography's "gyral bias" that failed to map connections at the "gyral bank." However, there could be spurious connections because tractography could not confirm innervation. For example, the connection probabilities between IFOF and insula regions (e.g., Pol1 and 52) were just due to IFOF passing by, and further histology validation is needed.

Comment 3:

Catani et al. did not associate SLF2 to the anterior segment of the arcuate fasciculus but only SLF3. Please correct or remove the sentence.

Responses:

The manuscript is corrected accordingly to remove SLF2.

Modifications Made:

Catani et al.20 showed SLF ~~I and~~ III as the anterior segment of the AF, which did not include SLF I.

REVIEWERS' COMMENTS

Reviewer #1 (Remarks to the Author):

The authors have addressed my comments and the manuscript is improved.

Reviewer #3 (Remarks to the Author):

I have no further comments